# STRUCTURE-PRESERVING OPERATOR LEARNING

## ABSTRACT

Learning complex dynamics driven by partial differential equations directly from data holds great promise for fast and accurate simulations of complex physical systems. In most cases, this problem can be formulated as an operator learning task, where one aims to learn the operator representing the physics of interest, which entails discretization of the continuous system. However, preserving key continuous properties at the discrete level, such as boundary conditions, and addressing physical systems with complex geometries is challenging for most existing approaches. We introduce a family of operator learning architectures, *structure-preserving operator networks* (SPONs), that allows to preserve key mathematical and physical properties of the continuous system by leveraging finite element (FE) discretizations of the input-output spaces. SPONs are encode-process-decode architectures that are end-to-end differentiable, where the encoder and decoder follows from the discretizations of the input-output spaces. SPONs can operate on complex geometries, enforce certain boundary conditions exactly, and offer theoretical guarantees. Our framework provides a flexible way of devising structure-preserving architectures tailored to specific applications, and offers an explicit trade-off between performance and efficiency, all thanks to the FE discretization of the input-output spaces. Additionally, we introduce a multigrid-inspired SPON architecture that yields improved performance at higher efficiency. Finally, we release a software to automate the design and training of SPON architectures.

## 1 INTRODUCTION

Partial differential equations (PDEs) underpin the modeling of many complex systems across science and engineering. However, traditional approaches such as the finite element method (FEM) are, in most cases, notoriously expensive, demand tailored solver configurations for each specific PDE, and cannot deal with scenarios where the underlying PDE modeling the system is unknown. Operator learning aims to address these limitations by approximating operators $\mathcal{G} : \mathcal{U} \to \mathcal{V}$ governed by PDEs, such as solution operators, directly from observational data (Boullé & Townsend, 2024; Kovachki et al., 2024), where $\mathcal{U}$ and $\mathcal{V}$ are infinite-dimensional function spaces. Operator learning has been successfully applied across different areas, including weather forecasting (Lam et al., 2023; Pathak et al., 2022; Kashinath et al., 2021) or continuum mechanics (You et al., 2022).

Several operator learning architectures have been proposed, ranging from graph networks, which leverage relational inductive biases (Pfaff et al., 2021; Brandstetter et al., 2022), to neural operators, which rely on discretizations of integral operators defined on infinite-dimensional spaces (Kovachki et al., 2023; Li et al., 2023), and physics-based architectures (Belbute-Peres et al., 2020; Li et al., 2024b), where the PDE serves as an inductive bias to encode physical prior knowledge. However, these techniques often consider pointwise discretizations of the input and output functions that discard the continuous mathematical structure of the function spaces considered, leading to inconsistencies between the continuous and discrete representation of the operator, which deteriorate the operator approximation (Bartolucci et al., 2023). Hence, structural properties at the continuous level, such as symmetries, boundary conditions, or conservation laws, may not be preserved at the discrete level. In addition, most existing approaches discard the topological information of the underlying domain, thereby restricting them to simple geometries or meshes (Li et al., 2021; 2024b).

In contrast, the finite-element method carries the topological information of the domain, and extensive literature has been devoted to structure-preserving discretizations, such as the finite element and exterior calculus framework (FEEC) (Arnold et al., 2006). While different approaches have been

proposed for enforcing constraints in operator learning (Jiang et al., 2024), a generic and consistent framework for designing operator learning architectures with structure-preserving spatial discretizations is lacking. Throughout this work, we denote structure-preserving discretizations as numerical methods that preserve, on the discrete level, key geometric, topological, and algebraic structures possessed by the original continuous system.

We introduce a family of structure-preserving operator learning architectures, called *structure-preserving operator networks* (SPON), that are expressed as encoder-processor-decoder models. The encoder and decoder result from the finite element (FE) discretization of the input-output spaces and the processor operates on FE degrees of freedom. As a result, SPON architectures are capable of naturally preserving key properties of the continuous operator thanks to FE discretizations, which can be tailored to specific scientific applications. Moreover, structure-preserving operator networks can operate on complex geometries and meshes, preserve certain boundary conditions exactly at the discrete level, while offering theoretical guarantees on the approximation error. We also demonstrate that our framework exhibits mesh-invariant capabilities and provide an explicit way to control the operator aliasing error via the discretization employed. Our framework achieves higher accuracy than several state-of-the-art architectures on a classical benchmark. Finally, we introduce a multigrid-based SPON that can be scaled to large problems and captures long-range information, while achieving high performance and accuracy.

**Main contributions.** Our main contributions are summarized as follows.

1. We propose a generic and flexible framework for operator learning that combines the finite element method with the encode-process-decode paradigm, allowing for the preservation of key properties of the continuous system at the discrete level using finite element discretizations tailored to the PDE of interest. Our framework can be used on complex meshes, for time-dependent and steady problems, and comes with theoretical guarantees.

2. We introduce a multigrid-based structure-preserving operator network (SPON-MG) that combines multilevel message passing GNNs with finite element mapping operators. SPON-MG achieves greater accuracy with significantly higher efficiency while greatly reducing the number of parameters needed, resulting in lower memory usage and improved latency.

We also release an open-source library interfacing with the Firedrake FE software (Ham et al., 2023) for building structure-preserving operator networks using state-of-the-art FE discretizations.

## 2 BACKGROUND AND RELATED WORK

**Neural operators.** Kovachki et al. (2023) introduced neural operators as infinite-dimensional generalizations of neural networks to discretize and approximate operators associated with PDEs. Several architectures have been proposed and all result from a specific parametrization of the integral kernel (Boullé & Townsend, 2024). Examples include Fourier neural operators (FNO) (Li et al., 2021), which employ Fourier convolutional kernels, DeepONet (Lu et al., 2021), that learns the mapping between Hilbert spaces to a finite-dimensional latent space using encoder-decoder architectures (Kovachki et al., 2024), and Boullé et al. (2022a) that learns Green's functions. Such approaches consider samples of the input-output functions at point values or on a tensor-product grid and often discard the intrinsic structures of the underlying continuous PDE systems, leading to aliasing errors (Bartolucci et al., 2023). In contrast, we consider a FEM discretization of the input-output spaces, which allows preserving key properties of the continuous spaces via the discretization, and are applicable to complex meshes. Notably, the aliasing error can be explicitly controlled by the choice of discretization.

**Operator learning and FEM.** Several related works explored connections between operator learning and the finite element method (FEM) (Cao, 2021; Franco et al., 2023; He et al., 2024; Lee et al., 2023; Xu et al., 2024). The MgNO introduced by He et al. (2024) uses a multigrid approach to discretize integral operators on simple geometries with tensor-product grids, and requires custom-defined convolution kernels to enforce specific boundary conditions. Franco et al. (2023) proposed a mesh-informed neural network for operator learning that uses a dense feedforward model along with a mesh that uses a pruning strategy to dismiss far points. Finally, Lee et al. (2023) considered the predictions of the neural operator at degrees of freedom, in cases where they coincide with the mesh vertex nodes, and used continuous Lagrange elements of degree one ($CG_1$).

**GNN simulators.** GNN-based methods have also been proposed for learning operators driven by PDEs. Pfaff et al. (2021) introduced an encode-process-decode GNN architecture (Battaglia et al., 2018) followed by a time integrator capable of learning mesh-based quantities. In (Brandstetter et al., 2022), a message passing neural PDE solver is considered to learn solution operators, along with a stabilization technique to train autoregressive operators. Finally, Belbute-Peres et al. (2020) embedded a differentiable CFD solver into a GNN to improve generalization. Our framework contrasts with these GNN architectures by using FEM discretizations to design a family of GNN architectures that preserve continuous structure of the operator at the discrete level, while being compatible with existing autoregressive techniques for time modeling.

**Physics-based approaches.** Different works explored the use of physics-based inductive biases for machine learning algorithms. Examples include the use of the PDE as a regularization term in the loss (Li et al., 2024b), constraining the architecture to enforce certain boundary conditions (Saad et al., 2023), or the design of neural networks that comply with thermodynamics principles (Hernández et al., 2021) or preserve structures of kinetic collision operators (Lee et al., 2024). In contrast, our approach relies on a FEM-based inductive bias that allows preserving mathematical properties of the continuous operator at the discrete level, offers theoretical guarantees, and allows tackling problems defined on complex geometries. Notably, the use of FEM discretizations facilitates the combination of complex physics-based inductive biases with SPON models (Bouziani et al., 2024).

## 3 METHOD

Our main motivation is to learn a (typically nonlinear) operator $\mathcal{G} : \mathcal{U} \to \mathcal{V}$ associated with a PDE (e.g., solution operator or inverse problem), where $\mathcal{U}$ and $\mathcal{V}$ are Hilbert spaces of functions defined on a bounded domain $\Omega \subset \mathbb{R}^d$ in spatial dimension $d \in \{1, 2, 3\}$. For simplicity, we consider $\mathcal{U}$ and $\mathcal{V}$ to be defined on the same domain $\Omega$, but different bounded domains and spatial dimensions may be considered. The spaces $\mathcal{U}$ and $\mathcal{V}$ arising from such problems are typically infinite-dimensional and need to be discretized. Similarly to FEM, we consider a mesh $\mathcal{M}$ of the domain $\Omega$, and two finite-dimensional spaces $\mathcal{U}_h$ and $\mathcal{V}_h$ arising from a suitable discretization of the spaces $\mathcal{U}$ and $\mathcal{V}$. We introduce a framework for designing operator learning architectures, which we refer to as *structure-preserving operator networks*, that approximate $\mathcal{G}$ on the discretized spaces $\mathcal{U}_h$ and $\mathcal{V}_h$.

### 3.1 STRUCTURE-PRESERVING OPERATOR NETWORK

We define a structure-preserving operator network (SPON) $\mathcal{S}_\theta$ between the finite-dimensional functions spaces $\mathcal{U}_h$ and $\mathcal{V}_h$ of dimensions $n$ and $m$ as

$$\mathcal{S}_\theta(f) = \mathcal{D} \circ \mathcal{P}_\theta \circ \mathcal{E}(f), \quad f \in \mathcal{U}_h, \tag{1}$$

where $\mathcal{E}$ and $\mathcal{D}$ denote the *encoder* and *decoder*, while $\mathcal{P}_\theta \colon \mathbb{R}^n \mapsto \mathbb{R}^m$ is a learnable model of parameters $\theta$, referred to as the *processor* (see Fig. 1).

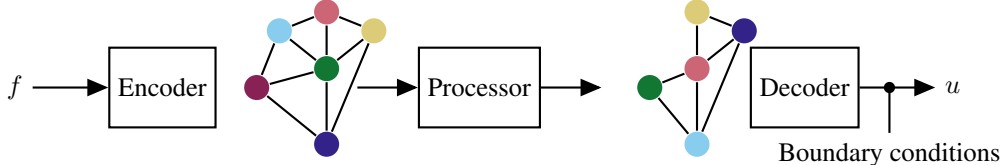

Figure 1: Schematic diagram of a structure-preserving operator network architecture.

**Encoder.** The encoder maps an input function $f \in \mathcal{U}_h$ to its degrees of freedom in the finite element space $\mathcal{U}_h = \text{span}(\varphi_1, \ldots, \varphi_n)$ as

$$\mathcal{E}(f) = (f_1, \ldots, f_n), \tag{2}$$

where $f_i = \langle f, \varphi_i \rangle$ denotes the Galerkin projection onto the $i$-th basis function $\varphi_i$.

**Decoder.** The decoder maps the predicted degrees of freedom in $\mathcal{V}_h$ to the reconstructed solution $u \in \mathcal{V}_h$ as

$$\mathcal{D}(u_1, \ldots, u_m) = u, \tag{3}$$

where $u(x) = \sum_{i=1}^{m} u_i \phi_i(x)$, for $x \in \Omega$, and with $(\phi_i)_{1 \leq i \leq m}$ a basis of $\mathcal{V}_h$.

The structure-preserving operator network framework combines the finite element method with the encode-process-decode paradigm. The finite element method can be seen as an encode-process-decode approach, with encoder $\mathcal{E}$ and decoder $\mathcal{D}$, and where the processor numerically solves the discretized system posed on the degrees of freedom (DoFs). On the other hand, classical encode-process-decode architectures (Battaglia et al., 2018) consider learnable models for the encoder, processor, and decoder. In contrast, only the processor $\mathcal{P}_\theta$ is learnable in our approach. In that sense, our framework can be seen as an operator learning approach over a structured latent space.

SPON architectures separate the concerns of the latent space representation, which relies on the finite element discretization, from the learning of the operator, which is delegated to the processor. The FEM-based encoder and decoder allow to leverage the rich literature on efficient structure-preserving finite element discretizations of PDE systems to preserve the structure of the input-output spaces. In particular, several key properties are naturally preserved for SPON architectures. This includes the preservation of the function space of the output, which is always $\mathcal{V}_h$ independently of $\mathcal{P}_\theta$. In fact, for conforming FE discretizations (Braess, 2001, Chapt. 2), the output also lies in $\mathcal{V}$ as in this is case we have $\mathcal{V}_h \subset \mathcal{V}$. Additionally, SPON architectures can impose boundary conditions such as Dirichlet exactly at the discrete level. Other mathematical and physical properties may be satisfied at the discrete level using structure-preserving FE discretizations (Arnold et al., 2006).

The choice of the discretized spaces is problem-specific, and different discretizations can be employed to incorporate prior information about the regularity of the input-output functions. Structure-preserving operator networks also inherit other FEM benefits, such as allowing arbitrary geometric decompositions, which facilitate the use of complex geometries, while offering an easy way to strike favorable trade-offs between training/inference time and accuracy through mesh refinement or higher-order spatial discretizations. One can, however, still provide pointwise values as inputs to the encoder and then use a suitable Galerkin projection to obtain the corresponding degrees of freedom.

**Relational inductive bias.** Since the spaces $\mathcal{U}_h$ and $\mathcal{V}_h$ are finite element spaces, the basis functions $(\varphi_i)_{1 \leq i \leq n}$ and $(\phi_i)_{1 \leq i \leq m}$ are typically piecewise polynomial functions with compact support, which results in sparse representations as most basis functions decouple, i.e., their supports do not intersect. More precisely, the value of a given function $f \in \mathcal{U}_h$ at a point $x \in \Omega$ depends only on the small set of basis functions whose supports contain $x$. This sparse representation induces a graph whose vertices correspond to the degrees of freedom $(f_i)_{1 \leq i \leq n}$ of $\mathcal{U}_h$, and where two vertices $f_i$ and $f_j$ admits an edge only if the corresponding basis functions $\varphi_i$ and $\varphi_j$ have overlapping support (see Appendices A.1 and A.2 for details). We use this graph representation as a relational inductive bias for the processor $\mathcal{P}_\theta$ that maps the graph associated with the degrees of freedom of $\mathcal{U}_h$ to the ones of $\mathcal{V}_h$, as illustrated in Fig. 1. This motivates the use of graph neural network architectures for $\mathcal{P}_\theta$.

**Boundary conditions.** The function spaces $\mathcal{U}$ and $\mathcal{V}$ may be equipped with boundary conditions. For example, $\mathcal{V} = H_0^1(\Omega)$ contains functions that vanish on the boundary $\partial\Omega$. These continuous structures need to be preserved at the discrete level. Structure-preserving operator networks automatically conserve such boundary conditions at the discrete level, independently of the choice of the processor $\mathcal{P}_\theta$. This is achieved by the decoder, as outlined in Fig. 1, which enforces boundary conditions directly on the degrees of freedom. This allows to output functions that satisfy boundary conditions such as Dirichlet conditions exactly (see Appendix A.3).

**Time-dependent operators.** While our framework primarily addresses the spatial discretization of operators between function spaces, SPON architectures can also be used for time-dependent problems. The SPON "function-to-function" interface provides a composable and flexible way to model time-dependent operators and can seamlessly be integrated with standard temporal modeling approaches, such as autoregressive modeling (Pfaff et al., 2021; Lam et al., 2023), neural operators (Li et al., 2021), or temporal bundling (Brandstetter et al., 2022).

**Zero-shot super resolution.** The structure-preserving operator network $\mathcal{S}_\theta$ produces a finite element function $u \in \mathcal{V}_h$, which can be evaluated at any point $x$ in the geometrical domain $\Omega$ via $u(x) = \sum_{i=1}^{m} u_i \phi_i(x)$, independently of the mesh and resolution it was trained on. This powerful property results from the FE discretization of $\mathcal{U}$ and $\mathcal{V}$ and holds even for complex geometries. Such a property is highly desirable to transfer solutions between different meshes and space discretizations, e.g., for zero-shot super resolution, and yields architectures that can operate across different

resolutions. Let $\mathcal{S}_\theta : \mathcal{U}_h \to \mathcal{V}_h$, we can construct the SPON interpolation operator $\hat{\mathcal{S}}_\theta : \mathcal{X}_h \to \mathcal{Y}_h$:

$$\hat{\mathcal{S}}_\theta = \mathcal{P} \circ \mathcal{S}_\theta \circ \mathcal{R}, \tag{4}$$

where $\mathcal{X}_h$ and $\mathcal{Y}_h$ are discrete spaces defined on a different resolution and/or different finite element discretization than $\mathcal{U}_h$ and $\mathcal{V}_h$, and with $\mathcal{R} : \mathcal{X}_h \to \mathcal{U}_h$ and $\mathcal{P} : \mathcal{V}_h \to \mathcal{Y}_h$ the corresponding finite element interpolation operators. In the zero-shot super resolution case, $\mathcal{R}$ and $\mathcal{P}$ are merely the restriction and prolongation operators. Note that $\hat{\mathcal{S}}_\theta$ and $\mathcal{S}_\theta$ have the same number of parameters.

## 3.2 A MULTIGRID-INSPIRED PROCESSOR

The latent graph representations of SPONs promote the use of GNN architectures for the processor. The corresponding graphs intrinsically depend on the mesh and the chosen finite element discretizations. For example, higher-resolution meshes or higher-order discretizations lead to larger graphs. However, the use of standard GNNs such as message-passing architectures on large graphs is doomed by significantly higher computational cost and information bottleneck, i.e., the model cannot capture long-range information since a GNN with $m$ layers can only capture information up to $m$ hops away. While increasing the number of message passings helps propagate information through the graph, it also increases the computational load and may lead to over-smoothing.

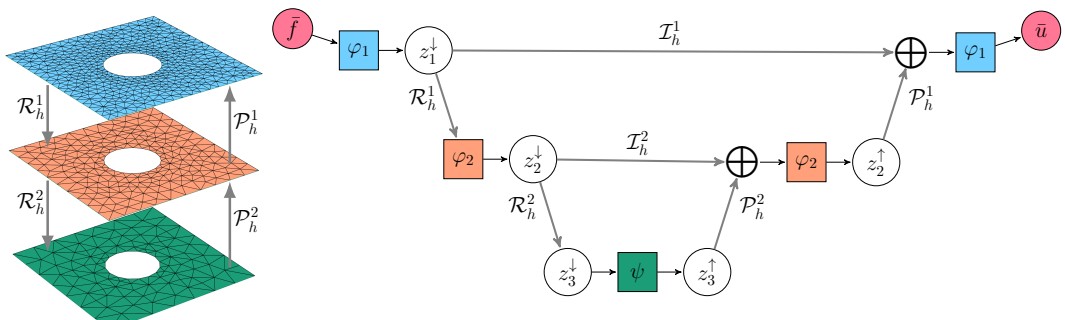

Figure 2: Diagram of the multigrid processor $\mathcal{P}_\theta^{MG}$ for 3 levels (right) with the corresponding mesh hierarchy (left). $\mathcal{P}_\theta^{MG}$ takes in $\bar{f}$, the input DoFs, and predicts $\bar{u}$, the DoFs of the output.

To address these limitations, we introduce a multigrid-inspired processor $\mathcal{P}_\theta^{MG}$ that operates on a hierarchy of meshes and function spaces to provide greater accuracy with higher efficiency. Our processor yields a structure-preserving operator network that can scale to highly resolved meshes and/or high-order discretizations, while efficiently capturing long-range dependencies between distant regions in the domain. Our multilevel processor combines lightweight message passing architectures $(\varphi_i)_{1 \le i \le N}$ at each level, which models information exchange at different length scales, with a larger graph-based architecture $\psi$ that facilitates the propagation of information at the coarse level. The computational load is delegated to the coarse model $\psi$, which performs message-passing updates that are significantly cheaper than the fine level, thereby increasing the computational efficiency. This processor contrasts with existing multiscale GNN approaches (Fortunato et al., 2022; Li et al., 2020; Lam et al., 2023), where each scale is only defined by a given mesh resolution. Our approach combines the mesh resolution with the FE discretization of the input-output spaces, associating each level with a pair of function spaces. It may be described as a *functional multilevel message passing* since the latent features across the different scales can all be associated with a given function in a known function space. Then, mapping latent features from one space to another is achieved using appropriate operators between the FE spaces.

We define $\mathcal{R}_h^i : \mathcal{U}_h^i \to \mathcal{U}_h^{i+1}$, $\mathcal{P}_h^i : \mathcal{V}_h^{i+1} \to \mathcal{V}_h^i$, and $\mathcal{I}_h^i : \mathcal{U}_h^i \to \mathcal{V}_h^i$, the *restriction*, *prolongation*, and *interpolation* operators, respectively. These operators are used across the processor architecture to map latent features. This is achieved through a sparse matrix-vector product, where the matrix results from the discretization of the operator, and the vector contains the latent degrees of freedom. This additional inductive bias drastically decreases the number of parameters needed, as the matrices do not have to be learned, unlike other approaches (Li et al., 2020), and reduces the model latency. Notably, the matrices induced by $(\mathcal{R}_h^i)_i$, $(\mathcal{P}_h^i)_i$, and $(\mathcal{I}_h^i)_i$ are sparse, which reduces memory and allows for larger batch sizes, leading to higher throughput. The architecture of $\mathcal{P}_\theta^{MG}$, outlined in

Fig. 2, is inspired by the V-cycle multigrid algorithm and multilevel message passing methods. It consists of a downward pass, a coarse update, and an upstream pass. See Appendix B for more details.

**Downward pass.** Let $2 \leq i \leq N - 1$, the downward pass is defined as $z_i^{\downarrow} = \varphi_i(\mathrm{R}^{i-1} z_{i-1}^{\downarrow})$, along with $z_1^{\downarrow} = \varphi_1(\bar{f})$ and $z_N^{\downarrow} = \mathrm{R}^{N-1} z_{N-1}^{\downarrow}$.

**Coarse update.** The coarse update $z_N^{\uparrow} = \psi(z_N^{\downarrow})$ consists of a forward pass through a messaging-passing-based model $\psi$ with linear encoding and decoding.

**Upward pass.** Let $1 \leq i \leq N - 1$, the upward pass is defined as $z_i^{\uparrow} = \varphi_i(I_{i-1} z_i^{\downarrow} \bigoplus \mathrm{P}^i z_{i+1}^{\uparrow})$, where $\bigoplus$ is a differentiable aggregator such as averaging, or a learnable linear combination. We then return $\bar{u} = z_1^{\uparrow}$.

### 3.3 Approximation error

This section provides an error estimate for the approximation of a Lipschitz continuous operator $\mathcal{G}$ by a structure-preserving operator network $\mathcal{S}_\theta$. Let $\Omega_1 \subset \mathbb{R}^{n_1}$ and $\Omega_2 \subset \mathbb{R}^{n_2}$ be two open bounded domains, $\mathcal{U}$ be a compact subset of a Hilbert space of functions $H^{k_1}(\Omega_1)$, for $k_1 \geq 0$, defined as $\mathcal{U} = \{f \in H^{k_1}(\Omega_1), \|f\| \leq 1\}$, and $\mathcal{V} = H^{k_2}(\Omega_2)$ for some $k_2 \geq 0$. Let $\mathcal{U}_{h_1}$ and $\mathcal{V}_{h_2}$ be two finite element spaces of functions defined on regular (in the sense of Brenner & Scott 2008, Def. 4.4.13) meshes $\mathcal{M}_{h_1}$ and $\mathcal{M}_{h_2}$ of $\Omega_1$ and $\Omega_2$, respectively, with polynomial degree $k_1$ and $k_2$, and maximum diameter $h_1$ and $h_2$. We assume that $\mathcal{U}_{h_1}$ and $\mathcal{V}_{h_2}$ are conforming finite element spaces (i.e., $\mathcal{U}_{h_1} \subset \mathcal{U}$ and $\mathcal{V}_{h_2} \subset \mathcal{V}$) satisfying the standard finite element hypotheses of Brenner & Scott (2008, Thm. 4.4.4). In addition, we denote by $P_{\mathcal{U}} : \mathcal{U} \to \mathcal{U}_{h_1}$ the Galerkin interpolation.

**Theorem 1** (Approximation bound). *Let $\mathcal{G} : H^{s_1}(\Omega_1) \to \mathcal{V}$ be a Lipschitz continuous operator for some $0 \leq s_1 \leq k_1$ and $0 < \epsilon < 1$. There exists a structure-preserving operator network $\mathcal{S}_\theta : \mathcal{U}_h \to \mathcal{V}_h \subset \mathcal{V}$ with a number of parameters bounded by*

$$|\theta| < C_1 \epsilon^{-C_2/h_1^{k_1 n_1}} (\log(1/\epsilon) + 1),$$

*such that for all $f \in \mathcal{U}$ and $0 \leq s_2 \leq k_2$,*

$$\|(\mathcal{G} - \mathcal{S}_\theta \circ P_{\mathcal{U}})(f)\|_{H^{s_2}(\Omega_2)} \leq C_3 \left( h_1^{k_1 - s_1} \|f\|_{H^{k_1}(\Omega_1)} + h_2^{k_2 - s_2} \|u\|_{H^{k_2}(\Omega_2)} \right) + \epsilon(h), \quad (5)$$

*where $u = \mathcal{G}(f)$, $C_1 > 0$ is a constant independent of $\epsilon$, and $C_2, C_3 > 0$ do not depend on $h_1, h_2, \epsilon$.*

The first two terms denote the finite element error on the input and output spaces with respect to $h_1$ and $h_2$, while the last term in Eq. (5) characterizes the quality of the neural network approximation as the number of parameters increases. The left-hand side in Eq. (5) can be seen as an operator aliasing error (Bartolucci et al., 2023), which can be explicitly controlled by the mesh resolution and the discretization of the input-output spaces. The proof of Theorem 1 is deferred to Appendix C and combines standard finite element bounds along with an error analysis similar to the ones derived by Kovachki et al. (2021); Lanthaler et al. (2022) for FNOs and DeepONets.

### 3.4 Software

We release an open-source software, `spon`, for designing structure-preserving operator networks that interface with the *Firedrake* (Ham et al., 2023), *PyTorch* (Paszke et al., 2019), and *physics-driven-ml* (Bouziani & Ham, 2023) packages. The `spon` package leverages the Firedrake API for specifying meshes, as well as a rich collection of finite element spaces, including elements such as Lagrange, discontinuous Galerkin, and Raviart-Thomas. The Firedrake interface also allows for the specification of complex boundary conditions or input functions from observable point data, which is convenient for certain operator learning applications. Our software automates the encoder and decoder of SPON architectures, which includes the construction of the latent graphs and boundary conditions support. Additionally, our package facilitates the construction of the mesh and function space hierarchies for multigrid processors, along with the restriction, prolongation, and interpolation operators for nested and non-nested meshes. Listing 1 outlines how SPON models can be implemented using our interface. Moreover, recent advances in differentiable programming (Bouziani et al., 2024; Bouziani & Ham, 2021) enable SPONs to be coupled with PDE constraints implemented in Firedrake.

```python
import firedrake as fd
import spon
...
# Define the discretized function spaces 𝒰ₕ and 𝒱ₕ
mesh = ...
U = fd.FunctionSpace(mesh, "CG", 1)
V = fd.FunctionSpace(mesh, "CG", 2)

# Define boundary conditions
bcs = fd.DirichletBC(V, Constant(0.), "on_boundary")

# Define the processor 𝒫_θ
processor = ...

# Define the structure-preserving operator network 𝒮_θ
S = spon.SPON(U, V, processor=processor, bcs=bcs)
```

Listing 1: Outline of the `spon` interface. A SPON model is defined in line 16, mapping from $\mathcal{U}_h$ to $\mathcal{V}_h$, where $\mathcal{U}_h$ (resp. $\mathcal{V}_h$) results from a continuous Lagrange discretization of degree 1 (resp. 2) as defined in line 6 (resp. line 7). Homogeneous Dirichlet boundary conditions are specified on the entire domain boundary in line 10.

## 4 NUMERICAL EXPERIMENTS

We evaluate the performance of our framework on several numerical examples pertaining to different physics and discretizations. We begin with a standard Poisson problem in Section 4.1 to demonstrate key properties of our framework and compare it with state-of-the-art architectures. In Section 4.2, we learn the time-dependent incompressible Navier-Stokes solution operator in a quasi-turbulent regime, on a complex geometry and with boundary conditions using a highly refined mesh. Additional details are provided in Appendix D, and further experiments are conducted in Appendix E.1.

### 4.1 POISSON'S EQUATION WITH STRONG BOUNDARY CONDITIONS

We consider a 2D Poisson equation on the unit square $\Omega = [0, 1]^2$ with a Dirichlet condition:

$$-\nabla^2 u = f \quad \text{in } \Omega, \qquad u = g \quad \text{on } \Gamma_D, \tag{6}$$

where $f \in H^1(\Omega)$, $g \in C^\infty(\Gamma_D)$, and $\Gamma_D$ is the top boundary of $\Omega$. We aim to approximate the corresponding solution operator $\mathcal{G} : H^1(\Omega) \to H^2(\Omega) \cap H^1_g(\Omega)$ that maps the source term $f$ to the solution $u$ of Eq. (6). The performance and efficiency of our approach are compared with FNO (Li et al., 2021) and DeepONet (Lu et al., 2021). Due to the requirements of FNO and DeepONet, we consider a uniform $n_x \times n_x$ grid, with different resolutions ranging from $n_x = 16$ to 256. To facilitate the comparison with benchmarks, we use the same discretization for the input and output spaces, namely a $CG_1$ finite element space. We consider two types of structure-preserving operator networks for this experiment: one with the single-level processor $\psi$ (see Eq. (10)), which we simply refer to as *SPON*, and the multigrid processor introduced in Section 3.2, which we denote by *SPON-MG*.

Our experiments demonstrate that our framework outperforms the benchmarks by a significant margin for both the single-level and multigrid architectures, as illustrated in Fig. 3a and Table 1. We also display the preservation of the Dirichlet boundary condition from the continuous output space at the discrete level, resulting in exact matching of the boundary condition on $\Gamma_D$ (cf. Table 1), unlike other methods. We evaluate the performance of our multigrid processor, which results in a model that surpasses the best benchmark we compare it with while having 3.5 times fewer parameters, illustrating the benefits of the multigrid FE structure we employed. The single-level *SPON* architecture stands out as the fastest approach. However, we note that *SPON-MG* is slower than *SPON* at this resolution due to the overhead induced by the mappings across levels. This is compensated at higher resolutions, where *SPON-MG* is significantly faster while massively reducing the number of parameters, as illustrated in Fig. 11a. We emphasize that our framework primarily aims to provide a flexible way to preserve continuous structure at the discrete level while allowing for arbitrary geometries, rather than to outperform existing architectures.

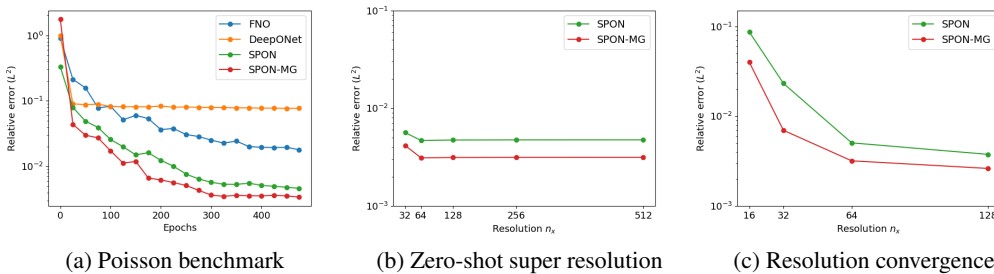

(a) Poisson benchmark
(b) Zero-shot super resolution
(c) Resolution convergence

Figure 3: **Left:** Relative errors across the epochs for different benchmarks (trained and tested at $n_x = 64$). **Middle:** Relative errors of *SPON* and *SPON-MG* trained at $n_x = 64$ and evaluated at different resolutions **Right:** Relative errors when trained and evaluated at different resolutions.

**Zero-shot super resolution.** We compare the performance of both SPON models trained at a fixed resolution and evaluated at different resolutions using the SPON interpolation operator, see Eq. (4). We observe in Fig. 3b that both models exhibit *mesh-invariance* capabilities similar to FNO (Li et al., 2021), i.e., their performance remains constant when evaluated on finer resolutions.

**Discretization dependence.** We compare the approximation error of both SPON models trained for different resolutions. We show that as the resolution becomes finer, the approximation error decreases, as outlined in Fig. 3c, which is in agreement with Theorem 1 since resolution and polynomial order are two ways by which the approximation error of structure-preserving operator networks can be improved. Note that the $CG_1$ approximation used to discretize the input-output spaces should result in a linear convergence rate with respect to $h$ ($=1/n_x$). However, this is not directly observable from Fig. 3c since the $\epsilon(h)$ term in Theorem 1 is different for each resolution $n_x$.

Table 1: Benchmarks on Poisson ($64 \times 64$ resolution for both training and testing).

| Method | Parameters | Epoch time | Relative $L^2-$error | $\|u - g\|_{L^2(\Gamma_D)} / \|g\|_{L^2(\Gamma_D)}$ |
|---|---|---|---|---|
| FNO | $1,200,225$ | $1.49s$ | $1.71 \times 10^{-2}$ | $1.85 \times 10^{-1}$ |
| DeepONet | $3,484,417$ | $2.38s$ | $8.09 \times 10^{-2}$ | $4.38 \times 10^{-1}$ |
| SPON | $3,568,270$ | **1.19s** | $4.52 \times 10^{-3}$ | **0** |
| SPON-MG | $338,827$ | $2.25s$ | $\mathbf{3.21 \times 10^{-3}}$ | **0** |

**Multigrid processor.** We compare the efficiency of *SPON* and *SPON-MG* at different resolutions in Appendix E.2. Our main finding is that *SPON-MG* achieves significantly higher efficiency above a certain resolution, while consistently requiring less parameters across all resolutions, thereby further improving latency (see Fig. 11).

## 4.2 FLUID FLOW PAST A CYLINDER

In this example, we aim to learn the time-forward operator $u(\cdot, t) \mapsto u(\cdot, t + \Delta t)$ associated with a classical cylinder flow benchmark (Jackson, 1987), for $t \geq 0$ and a timestep $\Delta t > 0$. The fluid velocity $u : \Omega \times [0, T] \to \mathbb{R}^2$ is governed by the incompressible Navier–Stokes equations:

$$\frac{\partial u}{\partial t} - \nabla \cdot \frac{2}{\text{Re}} \varepsilon(u) + (u \cdot \nabla)u + \nabla p = f \quad \text{in } \Omega \times (0, T],$$
$$\nabla \cdot u = 0 \quad \text{in } \Omega \times [0, T], \tag{7}$$

where $p$ is the pressure, $\Omega \subset \mathbb{R}^2$ is the computational domain, $\text{Re} = 200$ is the Reynolds number, and $\varepsilon(u) = (\nabla u + \nabla u^\top)/2$. We equip Eq. (7) with the following boundary conditions: $u = (1, 0)^\top$ on the upper and lower side of $\Omega$, a homogeneous Dirichlet boundary condition on the obstacle, and a "do-nothing" condition on the right side of $\Omega$ (see Fig. 4). We consider fluid velocities up to $T = 30$ and $\Delta t = 2$, which include laminar flow as well as vortex structures. The dataset is formed by considering different random inflow conditions on the left boundary. The discrete space $\mathcal{U}_h$ ($= \mathcal{V}_h$) is derived using the standard Taylor-Hood finite elements, a stable element for

the Stokes equation (Taylor & Hood, 1973). We consider an unstructured mesh to represent the non-trivial geometry $\Omega$. The discrete spaces result in a graph of about $40k$ nodes and $800k$ edges, highlighting the computational challenge of this example. Additionally, the different source terms, the autoregressive nature of the network, and the change in physics occurring along the way, from laminar to periodic with complex von Kármán vortex street patterns, present additional challenges for the network to capture. We consider the multigrid processor $\mathcal{P}_\theta^{MG}$ for our SPON architecture

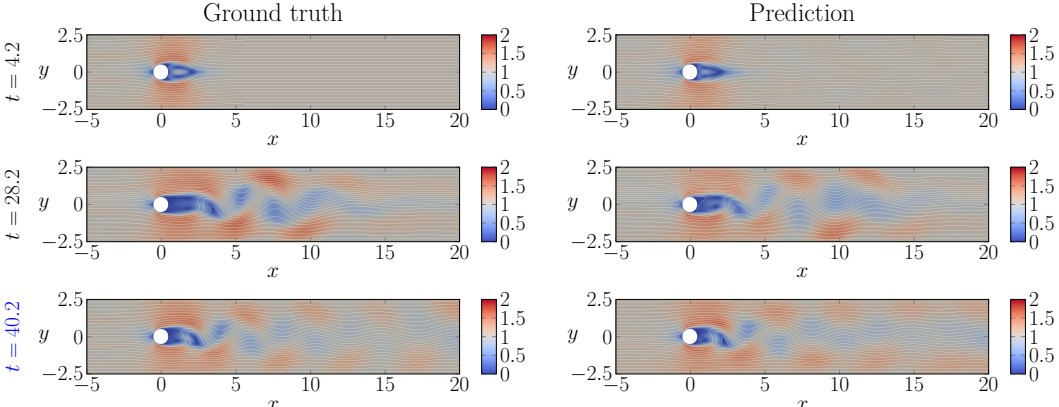

Figure 4: **Left:** Exact fluid velocity flow and magnitude from a random source in the test set (left boundary condition) at different time steps in the simulation. **Right:** Predicted velocity solution from the same initial condition at $t = 2.2$. The bottom row (highlighted in blue) is an extrapolation test as the time step has not been observed in training.

and generate a hierarchy of non-uniform meshes, as shown in Fig. 7. Our model inherently satisfies the strong boundary conditions imposed on the obstacle and on the upper and lower boundaries of $\Omega$ exactly. The one-step modeling of the velocity (Pfaff et al., 2021) allows generating long trajectories at inference time via iterative application of the model. However, it makes the task more difficult as such operators are prone to error accumulations. For training, we use a divergence-free regularization on the loss function (Bouziani et al., 2024), which can be interpreted as an incompressibility constraint.

Despite a relatively large timestep that intentionally misses fast-scale dynamics, we observe a good agreement between the ground truth and predicted solution, even at the last time step corresponding to a high number of model rollouts, as illustrated in Fig. 4. We achieve a relative $L^2$-error of $1.3 \times 10^{-1}$ on the test set ($7.1 \times 10^{-2}$ on the training set). Moreover, we observe relatively robust extrapolation capabilities of the model, demonstrating low errors on time steps not seen during training, see Fig. 8 and Fig. 9. The training and inference time of our architecture is dominated by the message passing updates occurring at the finest level in $\mathcal{P}_\theta^{MG}$. This can be attributed to the high number of nodes at that level and can be mitigated by reducing the number of message-passing layers specifically for that level or by using standard approaches for handling GNNs on large graphs. We highlight that this problem is highly challenging for most operator learning architectures due to the complex geometry with unstructured data points and long-term integration.

## 5 CONCLUSIONS

Structure-preserving operator networks provide a generic and flexible framework for learning operators modeling complex physical systems while preserving important properties of the system of interest at the discrete level. Our approach leverages the rich literature on finite element discretizations and can be tailored to specific physics. SPONs can operate on complex geometries and meshes, can be used in conjunction with time-dependent approaches, and demonstrate state-of-the-art performance with mesh-invariance capabilities. Structure-preserving operator networks come with theoretical guarantees and have an approximation error that can be explicitly reduced through discretization. Our multigrid processor facilitates scaling to larger problems and demonstrates greater performance with higher efficiency, while massively reducing the number of parameters. The FEM structure of our framework can be exploited to develop SPON variants that incorporate physical prior knowledge, which may improve generalization and increase accuracy.

**Limitations.** The finite element method is naturally suited for problems with spatial dimension $d \leq 3$. Our primary motivation is to address these problems. The application of SPONs to higher-dimensional problems can result in dense graphs, making the naive implementation of GNN-based processors prohibitively expensive. However, techniques such as sampling (Hamilton et al., 2017) or similar methods can be employed to mitigate this issue.

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

# Appendix

## TABLE OF CONTENTS

## A  STRUCTURE-PRESERVING OPERATOR NETWORKS

### A.1  FINITE ELEMENT DISCRETIZATION

The finite element method (FEM) is a numerical method to approximate the solution of partial differential equations. Given that PDE systems are naturally posed on infinite-dimensional spaces, discretization is required to solve these systems on a computer. Given that this work lies at the intersection between machine learning and numerical PDE solvers, we provide some background material on the finite element discretization for the readers not familiar with these methods.

Let $\mathcal{V}$ be an infinite-dimensional function space defined on a bounded domain $\Omega \subset \mathbb{R}^d$. The conforming finite element approach aims to approximate a solution $u \in \mathcal{V}$ to a given PDE on a finite-dimensional subspace $\mathcal{V}_h \subset \mathcal{V}$, characterized by a basis $(\phi_i)_{1 \le i \le N}$, where $N = \dim(\mathcal{V}_h)$. The approximation $u_h$ to $u$ can be written as $u_h = \sum_{i=1}^{N} u_i \phi_i$, and is determined by the coefficients $(u_i)_{1 \le i \le N} \in \mathbb{R}^N$, referred to as *degrees of freedom*, for a given choice of basis functions.

In a classical FEM context, the degrees of freedom are computed by solving the finite-dimensional system resulting from the discretization of the PDE of interest. In the proposed structure-preserving operator learning framework, the degrees of freedom are obtained by a structure-preserving operator network $\mathcal{S}_\theta$. Notably, SPONs are a tool to approximate a solution $u$ defined as $u = \mathcal{G}(f)$, for some operator $\mathcal{G}$ and parameter $f$, and are therefore not limited to approximate the solutions of PDE systems. Additionally, in our case, the finite element discretization is used for $u$, but also to discretize the input parameter $f$.

The finite element method originates from the idea of partitioning the computational domain $\Omega$, where the PDE is posed, into a collection of subdomains, or cells. More specifically, let $\mathcal{T}_h$ be a tessellation of $\Omega$ into *elements* defined as $\mathcal{T}_h := \{K_i\}_i$ such that $K_i \subset \Omega$ and $\overline{\Omega} = \cup_i K_i$, the interiors of $K_i$ and $K_j$ are disjoint for all $i \ne j$. $\mathcal{T}_h$ is referred to as a *mesh* of $\Omega$ as its edges and vertices form

```python
import firedrake as fd

# Define the mesh
mesh = fd.UnitSquareMesh(50, 50)
# Lagrange discretization
family = "CG"
# Polynomial degree
degree = 1
# Define the finite element space V_h
V = fd.FunctionSpace(mesh, family, degree)
```

Listing 2: Outline of the Firedrake interface for defining a finite element space with continuous piecewise polynomial of degree one on a unit square.

a mesh. The choice of basis functions is another crucial aspect in the finite element method and pertains to the chosen finite element discretization. The vast literature on the finite element method has led to a plethora of choices of discretization. In practice, numerical analysts consider tailored discretizations for each specific PDE system arising across science and engineering, such as those in electromagnetism, fluid dynamics, and elasticity. Structure-preserving discretizations have also been proposed to conserve mathematical and physical properties of certain PDEs, as outlined in Arnold & Logg (2014).

Finite element discretizations rely on a choice of basis functions with a small support adapted to the tessellation of the domain. This idea, introduced in Courant (1943), was motivated by the fact that the product of such a basis function with most of the other basis function vanishes, leading to sparse linear systems that can be solved efficiently. For more details about the historical developments of the finite element method, we refer to Gander & Wanner (2012). The small support assumption of the finite element discretization implies that each degree of freedom $u_i$ interacts with the small number of degrees of freedom whose basis functions have overlapping supports with $\phi_i$. This leads to a sparse graph representation of the degrees of freedom that is leveraged by our framework (cf. Appendix A.2 for more details).

A popular choice of discretization is the continuous Lagrange finite element discretization. In this case, the finite element space $\mathcal{V}_h$, also referred to as $\mathrm{CG}_k$, is chosen to be the space of continuous piecewise polynomials of degree $k$ defined as $\mathcal{V}_h = \{v \in C(\overline{\Omega}) \mid v_{|K_i} \in \mathbb{P}_k, \forall K_i \in \mathcal{T}_h\}$. The polynomial degree $k$ of the polynomials offers a trade-off between the computational efficiency of the numerical method and its accuracy, higher-order polynomials leading to more accurate approximations at the detriment of a higher computational cost. However, the efficient implementation of arbitrary finite element spaces such as $\mathrm{CG}_k$ can be a tedious task. The `spon` package, released with this paper, interfaces with the Firedrake FEM software (Ham et al., 2023) to automate the construction of a rich set of finite element spaces, which relies on code generation for high performance. Listing 2 illustrates how Lagrange finite element spaces can be simply defined in Firedrake. Finally, other discretizations may be considered, such as discontinuous Galerkin (Arnold et al., 2002) or the Raviart-Thomas elements (Raviart & Thomas, 2006), by changing the family keyword argument in line 6 of Listing 2.

### A.2 LATENT GRAPH REPRESENTATION

The SPON encoder maps finite element functions to their degrees of freedom and constructs a graph based on the sparsity of the underlying discretization. Let $\mathcal{V}_h$ be a finite element space discretizing an appropriate infinite-dimensional space $\mathcal{V}$ and let $(\phi_i)_i$ be the basis functions of $\mathcal{V}_h$. For every $u \in \mathcal{V}_h$, we have $u = \sum_i u_i \phi_i$, where $(u_i)_i$ denote the degrees of freedom of $\mathcal{V}_h$. We define the SPON latent graph of $\mathcal{V}_h$ as $\mathrm{G}_{\mathcal{V}_h} := (\mathrm{V}, \mathrm{E})$, where V is the set of vertices and E denotes the edges of the graph. The nodes of $\mathrm{G}_{\mathcal{V}_h}$ correspond to the degrees of freedom of $\mathcal{V}_h$ and the edges are defined by the sparsity of the discretization, which results from the fact that the basis functions $(\phi_i)$ all have a compact support. More specifically, we have:

$$\mathrm{V} := \{u_i\}_{1 \leq i \leq |\mathcal{V}_h|},$$

$$\mathrm{E} := \left\{ (i, j), \text{ where } i \text{ and } j \text{ are such that } \int_{\Omega} \phi_i \phi_j \, \mathrm{d}x \neq 0 \right\}. \tag{8}$$

Here, two nodes are connected by an edge if the corresponding basis functions have a spatial overlap. However, other criteria may be considered, leading to denser or sparser graph representations.

The released `spon` package automates the construction of the graph defined in Eq. (8). The construction of the edges is achieved by performing the finite element assembly of the mass matrix $M$, where $M_{ij} = \int_\Omega \phi_i \phi_j \, \mathrm{d}x$. We assemble $M$ and retrieves the column and row pairs of indices of its nonzero coefficients, which yields the adjacency list as defined in Eq. (8). The matrix assembly is achieved by the Firedrake finite element software (Ham et al., 2023), which relies on low-level generated code for high efficiency. We illustrate the latent graph induced by continuous Lagrange finite elements of order 1 and 2 on a square uniform and structured mesh with triangles in Fig. 5.

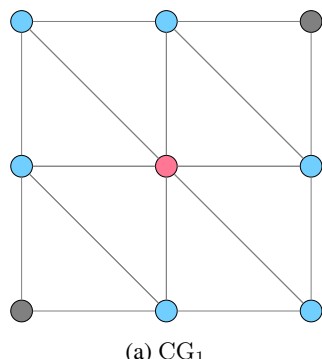 

(a) CG$_1$           (b) CG$_2$

Figure 5: Diagram illustrating the graph induced by a continuous Lagrange finite element of order 1 (CG$_1$) (left) and order 2 (CG$_2$) (right) on a square uniform and structured mesh with triangles. The neighbors (blue) of a given node (red) are illustrated. The latent graph's edges can be derived by finding the neighbors of each DoF using Eq. (8). Note that the graph's edges do not necessarily coincide with the mesh edges. In this example, for the CG$_1$ (resp. CG$_2$) discretization, each node has at most 7 (resp. 19) edges in the latent graph.

Our approach generalizes naturally for vector, tensor, and mixed function spaces. As an example, when solving the incompressible Navier–Stokes equations using the finite element method, a common discretization approach consists of solving for $w = (u_x, u_y, p)$, where $u = [u_x, u_y]$ is the velocity field and $p$ the pressure field. The velocity field is typically discretized using a continuous Galerkin finite element space of degree two, while the pressure field is discretized using a continuous Galerkin finite element space of degree one. Here, one would extract the sparsity pattern of the matrix $M$ defined as

$$M_{ij} = \int_\Omega w_i w_j^\top \, \mathrm{d}x = \int_\Omega \begin{bmatrix} u_{xi} u_{xj} & u_{xi} u_{yj} & u_{xi} p_j \\ u_{yi} u_{xj} & u_{yi} u_{yj} & u_{yi} p_j \\ p_i u_{xj} & p_i u_{yj} & p_i p_j \end{bmatrix} \, \mathrm{d}x.$$

In this case, the latent graph would have the degrees of freedom of $p$ connected to the degrees of freedom of $u$ and the degrees of freedom of the horizontal component of the velocity connected to the degrees of freedom of the vertical component of the velocity.

### A.3 BOUNDARY CONDITIONS SUPPORT

The decoder enforces Dirichlet boundary conditions strongly by assigning the degrees of freedom associated with the boundary to impose the boundary condition of interest exactly at the discrete level. Other boundary conditions, such as Neumann or Robin conditions, as well as global linear constraints, may be enforced similarly by solving a sparse system of equations on the boundary nodes to find the determine the values to enforce. This process may become computationally expensive in a training setting and may require a trade-off between accurately enforcing such boundary conditions and maintaining computationally efficiency. In such cases, a penalty term can be added in the loss function for a weak imposition of boundary conditions. Such explorations are left for future work.

# B  MULTIGRID-BASED MODEL

This section provides more details on the multigrid-based processor architecture introduced in Section 3.2. Another motivation for multigrid-based architectures comes from the recent theoretical studies on sample complexity for operator learning (Boullé & Townsend, 2023; Boullé et al., 2022b; 2023; 2024; de Hoop et al., 2023), which exploit regularity structure of partial differential operators (Bebendorf & Hackbusch, 2003) to show that one does not need many training data in operator learning to achieve a good approximation error. These results rely on a hierarchical decomposition of the spatial domain ($H$-matrix) (Bebendorf, 2008) to approximate short and long range interactions at different scales, similarly to the multigrid method.

## B.1  FUNCTION SPACE HIERARCHY AND MAPPING OPERATORS

Let $\mathcal{M}^1, \ldots, \mathcal{M}^N$ be a hierarchy of meshes with varying resolution, and $\mathcal{U}_h^1, \ldots, \mathcal{U}_h^N, \mathcal{V}_h^1, \ldots, \mathcal{V}_h^N$ be the corresponding input-output function spaces, where $\mathcal{U}_h^1 = \mathcal{U}_h$ and $\mathcal{V}_h^1 = \mathcal{V}_h$. The mesh hierarchy typically assumes nested meshes but non-nested meshes are also supported by our framework. We construct the mesh hierarchy in Firedrake using the procedure given in Listing 3.

The multigrid processor $\mathcal{P}_\theta^{MG} : \mathbb{R}^n \to \mathbb{R}^m$, introduced in Section 3.2, where $n = \dim(\mathcal{U}_h)$ and $m = \dim(\mathcal{V}_h)$ involves the restriction $\mathcal{R}_h^i : \mathcal{U}_h^i \to \mathcal{U}_h^{i+1}$ and prolongation $\mathcal{P}_h^i : \mathcal{V}_h^{i+1} \to \mathcal{V}_h^i$ operators, for $1 \le i \le N-1$, and the interpolation operator $\mathcal{I}_h^i : \mathcal{U}_h^i \to \mathcal{V}_h^i$, for $1 \le i \le N$. These are the classical finite element multigrid operators, often referred to as the "fine-to-coarse" and "coarse-to-fine" operators (Brenner & Scott, 2008, Sec. 6.3), along with a standard finite element interpolation operator. These operators map between finite element spaces and produce finite element functions as outputs. On the other hand, the multigrid processor only acts on the degrees of freedom. Given that these finite element operators are linear, they induce linear matrices that directly map the degrees of freedom in the input space to the degrees of freedom in the output space. More precisely, our processor uses the restriction, prolongation, and interpolation matrices $\mathrm{R}^i \in \mathbb{R}^{n_{i+1} \times n_i}$, $\mathrm{P}^i \in \mathbb{R}^{m_i \times m_{i+1}}$, and $\mathrm{I}^i \in \mathbb{R}^{m_i \times n_i}$, where $n_i = \dim(\mathcal{U}_h^i)$ and $m_i = \dim(\mathcal{V}_h^i)$. Notably, these matrices are all sparse and defined as follows:

$$\mathcal{R}_h^i(u_h^i) = \sum_{j=1}^{n_{i+1}} \left( \mathrm{R}^i u_j^i \right) \phi_j^{i+1} \in \mathcal{U}_h^{i+1}, \quad \mathcal{P}_h^i(v_h^{i+1}) = \sum_{j=1}^{m_i} \left( \mathrm{P}^i v_j^{i+1} \right) \psi_j^i \in \mathcal{V}_h^i,$$

$$\mathcal{I}_h^i(u_h^i) = \sum_{j=1}^{m_i} \left( \mathrm{I}^i u_j^i \right) \psi_j^i \in \mathcal{V}_h^i,$$

where $(\phi_j^i)_j$ and $(\psi_j^i)_j$ are the basis functions of the spaces $\mathcal{U}_h^i$ and $\mathcal{V}_h^i$, respectively, and $(u_j^i)_j$ are the degrees of freedom of $u_h^i \in \mathcal{U}_h^i$ and $(v_j^i)_j$ the degrees of freedom of $v_h^i \in \mathcal{V}_h^i$.

```python
import firedrake as fd

# Define a coarse mesh (M₁)
coarse_mesh = fd.UnitSquareMesh(32, 32)

# Define a hierarchy of meshes with 3 refinements (N = 4)
hierarchy = fd.MeshHierarchy(coarse_mesh, 3)

# Define the space hierarchy for CG₁ elements
V_spaces = [fd.FunctionSpace(m, "CG", 1) for m in hierarchy]
```

Listing 3: Outline of the Firedrake interface for defining a mesh hierarchy $\mathcal{M}^1, \ldots, \mathcal{M}^4$ and the corresponding corresponding finite element spaces $\mathcal{V}_h^i$, using a $\mathrm{CG}_1$ discretization and where the coarse mesh $\mathcal{M}_1$ is a unit square with 32 cells in each direction.

To speed up training and inference, we assemble these sparse matrices in Firedrake offline, and convert them to PyTorch sparse tensors. The restriction, prolongation, and interpolation operations, occurring across the architecture of $\mathcal{P}_\theta^{MG}$ during training and inference, are then performed via sparse matrix-vector products. Finally, it is worth noting that the restriction and prolongation operators

facilitate the inference of structure-preserving operator networks at different mesh resolution than the training resolution, as illustrated in Eq. (4).

## B.2 Message-passing models

The multigrid processor $\mathcal{P}_\theta^{MG}$ mainly relies on the learnable $(\varphi_i)_{1 \le i \le N}$ and $\psi$ message-passing-based architectures across the $N$ levels of the hierarchy. Each $\varphi_i$ is a lightweight model associated with the $i$-th level in the hierarchy and is used for both latent features on $\mathcal{U}_h^i$ and $\mathcal{V}_h^i$. These models are agnostic of the number of degrees of freedom and consist of $M$ identical message passing blocks $\phi$ with distinct sets of parameters. The $M$ blocks are combined in a pipeline manner with residual connections.

More precisely, let $H^0 = [h_1, \ldots, h_{n_{\text{DoFs}}}]^\top$ be a global feature vector containing the node features $(h_i)_{1 \le i \le n_{\text{DoFs}}}$, where $n_{\text{DoFs}}$ is the number of nodes in the graph, i.e., the number of degrees of freedom of the finite element space associated with the latent graph. For $1 \le i \le N$, we define $\varphi_i(H^0) = H^M$, where $H^M$ results from the following iterative procedure:

$$H^{n+1} = H^n + \alpha\,\phi(H^n),$$

for $0 \le n \le M-1$ and $\alpha > 0$. For our experiments, we consider $\alpha = \frac{1}{M}$. Here, $\phi$ is a message passing architecture defined by the following update.

$$\text{Edge } j \to i \text{ message update}: \quad m_{ij} = \phi_e\left(h_i, h_j - h_i\right),$$

$$\text{Node } i \text{ update}: \qquad\qquad h_i = \phi_v\left(h_i, \frac{1}{|\mathcal{N}(i)|}\sum_{j \in \mathcal{N}(i)} m_{ij}\right), \tag{9}$$

where $\mathcal{N}(i)$ denotes the neighborhood of the node feature $h_i$, and $\phi_e$ and $\phi_v$ are MLPs with "swish" activation functions (Ramachandran et al., 2017). We consider 4 linear layers for both MLPs. It is worth noting that the number of parameters of the message passing architectures does not depend on the number of degrees of freedom and therefore remains constant across the different hierarchy levels.

The $(\varphi_i)_{1 \le i \le N}$ models can only propagate information $M$ hops away at each level and are not enough to capture long-range dependencies. We employ a coarse model $\psi$ that allows efficient global exchange of information throughout the domain. $\psi$ operates at the coarser level and therefore enables node updates that are significantly cheaper than at the finer levels. The coarser model $\psi$ can also be seen as an encode-process-decode model with linear encoding and decoding that allow global exchange of information across all the DoFs. The processor of $\psi$ comprises message-passing updates via $\varphi_N$ on the graph associated with $\mathcal{U}_h^N$, followed by an interpolation to $\mathcal{V}_h^N$, and finally message-passing layers via $\varphi_N$ over the graph associated with $\mathcal{V}_h^N$. More specifically, the coarse model can be defined as:

$$\psi := W_{\mathcal{V}_h^N} \circ \varphi_N \circ \mathcal{I}_h^N \circ \varphi_N \circ W_{\mathcal{U}_h^N}, \tag{10}$$

with $\varphi_N$ the message-passing architecture defined in Eq. (9), $\mathcal{I}_h^N$ the interpolation operator between $\mathcal{U}_h^N$ and $\mathcal{V}_h^N$, and where $W_{\mathcal{U}_h^N} \in \mathbb{R}^{n \times n}$ and $W_{\mathcal{V}_h^N} \in \mathbb{R}^{m \times m}$ are learnable matrices acting over the degrees of freedom of $\mathcal{U}_h^N$ and $\mathcal{V}_h^N$, respectively, for $n = \dim(\mathcal{U}_h^N)$ and $m = \dim(\mathcal{V}_h^N)$. Notably, these linear layers dominate the number of parameters of the processor $\mathcal{P}_\theta^{MG}$ with $n^2$ and $m^2$ parameters, respectively.

To cut down the number of parameters, reduce the memory footprint, and improve latency, we consider a low-rank approximation of the matrices $W_{\mathcal{U}_h^N}$ and $W_{\mathcal{V}_h^N}$ with a compression factor of $k$. That is, we define

$$W_{\mathcal{U}_h^N} = W_{\mathcal{U}_h^N}^\uparrow W_{\mathcal{U}_h^N}^\downarrow, \quad W_{\mathcal{V}_h^N} = W_{\mathcal{V}_h^N}^\uparrow W_{\mathcal{V}_h^N}^\downarrow, \tag{11}$$

with $W_{\mathcal{U}_h^N}^\uparrow \in \mathbb{R}^{n \times n/k}$, $W_{\mathcal{U}_h^N}^\downarrow \in \mathbb{R}^{n/k \times n}$, reducing the number of parameters to $2n^2/k$, and $W_{\mathcal{V}_h^N}^\uparrow \in \mathbb{R}^{m \times m/k}$, $W_{\mathcal{V}_h^N}^\downarrow \in \mathbb{R}^{m/k \times m}$, reducing the number of parameters to $2m^2/k$.

## C  APPROXIMATION THEORY PROOF

Before proving Theorem 1, we introduce some necessary notations. Following Fig. 6, let $P_{\mathcal{U}}$, $P_{\mathcal{V}}$ denote the Galerkin interpolation operators onto the finite-dimensional space $\mathcal{U}_{h_1}$ and $\mathcal{V}_{h_2}$, $I_{\mathcal{U}}$ : $\mathcal{U}_{h_1} \to \mathcal{U}$ and $I_{\mathcal{V}} : \mathcal{V}_{h_2} \to \mathcal{V}$ the injection operators. Let $\mathcal{E}_{\mathcal{U}_{h_1}} : \mathcal{U}_{h_1} \to K \subset \mathbb{R}^n$, $\mathcal{D}_{\mathcal{V}_{h_2}} : \mathbb{R}^m \to \mathcal{V}_{h_2}$ be the structure-preserving encoder and decoder, and $\mathcal{P}_\theta : \mathbb{R}^n \to \mathbb{R}^m$ be the processor. Here, $K$ is the compact image of $\mathcal{U}_{h_1}$ under $\mathcal{E}_{\mathcal{U}_{h_1}}$, which is compact since $\mathcal{U}$ is compact. Moreover, we denote by $\hat{\mathcal{G}} = P_{\mathcal{V}} \circ \mathcal{G} \circ I_{\mathcal{U}}$ the finite element approximation of $\mathcal{G}$. We recall the standard finite element error estimate (Brenner & Scott, 2008, Eq. 4.4.28) that will be used extensively in the proof of Theorem 1:

$$\|f - P_{\mathcal{U}}(f)\|_{H^{s_1}(\Omega_1)} \leq C_1 h^{k_1 - s_1} \|f\|_{H^{k_1}(\Omega_1)}, \quad f \in \mathcal{U},$$
$$\|u - P_{\mathcal{V}}(u)\|_{H^{s_2}(\Omega_2)} \leq C_2 h^{k_2 - s_2} \|u\|_{H^{k_2}(\Omega_2)}, \quad u \in \mathcal{V},$$
$$(12)$$

for some constant $C_1, C_2 > 0$, $0 \leq s_1 \leq k_1$, and $0 \leq s_2 \leq k_2$.

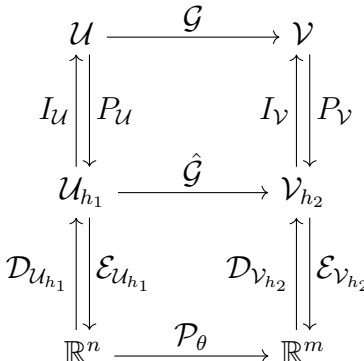

Figure 6: Diagram describing the operators used in the proof of Theorem 1.

*Proof of Theorem 1.* First, we remark that the function $\mathcal{E}_{\mathcal{V}_{h_2}} \circ \hat{G} \circ \mathcal{D}_{\mathcal{U}_{h_1}} : K \subset \mathbb{R}^n \to \mathbb{R}^m$ is Lipschitz continuous (a similar argument can be found in Lanthaler et al. 2022, Rem. 3.2). Moreover, we can embed $K$ into the hypercube $[-M, M]^n$ for sufficiently large $M > 0$. Let $\epsilon \in (0, 1)$, then by Yarotsky (2017, Thm. 1), there exists a ReLU neural network $\mathcal{P}_\theta : \mathbb{R}^m \to \mathbb{R}^n$ with $|\theta| \leq C_1 \epsilon^{-n}(\log(1/\epsilon) + 1)$ weights such that

$$\sup_{x \in K} \|\mathcal{E}_{\mathcal{V}_{h_2}} \circ \hat{G} \circ \mathcal{D}_{\mathcal{U}_{h_1}}(x) - \mathcal{P}_\theta(x)\|_{\ell^2(\mathbb{R}^m)} \leq \epsilon, \qquad (13)$$

where $C_1 > 0$ is a constant independent of $\epsilon$. Moreover, since $\mathcal{U}_{h_1}$ is a finite element space defined on a regular mesh, we have that $n = \dim(\mathcal{U}_{h_1}) \leq C_2 / h_1^{k_1 n_1}$ for some constant $C_2 > 0$.

Now, let $f \in \mathcal{U}$, $s_1 \leq k_1$, $s_2 \leq k_2$, and denote $f_{h_1} = P_{\mathcal{U}}(f)$. Then, the approximation error $\|(\mathcal{G} - \mathcal{S}_\theta \circ P_{\mathcal{U}})(f)\|_{H^{s_2}(\Omega_2)}$ can be expressed using triangular inequality in terms of the approximation error of the input-output spaces and the processor as

$$\|(\mathcal{G} - \mathcal{S}_\theta \circ P_{\mathcal{U}})(f)\|_{H^{s_2}(\Omega_2)} \leq \underbrace{\|\mathcal{G}(f) - \mathcal{G}(f_{h_1})\|_{H^{s_2}(\Omega_2)}}_{(A)} + \underbrace{\|\mathcal{G}(f_{h_1}) - \hat{\mathcal{G}}(f_{h_1})\|_{H^{s_2}(\Omega_2)}}_{(B)}$$
$$+ \underbrace{\|\hat{\mathcal{G}}(f_{h_1}) - \mathcal{S}_\theta(f_{h_1})\|_{H^{s_2}(\Omega_2)}}_{(C)},$$

where $\hat{\mathcal{G}} := P_{\mathcal{V}} \circ \mathcal{G} \circ I_{\mathcal{U}}$ is the finite element approximation of $\mathcal{G}$. Here, $(A)$ (resp. $(B)$) corresponds to the finite element approximation error of the input space $\mathcal{U}$ (resp. output space $\mathcal{V}$), and $(C)$ is the neural approximation error. We bound the three terms independently.

The term $(A)$ is bounded using the Lipschitz continuity of $G$ from $H^{s_1}(\Omega_1) \to H^{s_2}(\Omega_2)$ (as $\mathcal{V} \subset H^{s_2}(\Omega_2)$) and the finite element approximation bound in the input space $\mathcal{U}$ (Brenner & Scott, 2008, Eq. 4.4.28) as

$$\|\mathcal{G}(f) - \mathcal{G}(f_{h_1})\| \leq \text{Lip}(\mathcal{G})\|f - f_{h_1}\|_{H^{s_1}(\Omega_1)} \leq \text{Lip}(\mathcal{G})C_1 h_1^{k_1 - s_1}\|f\|_{H^{k_1}(\Omega_1)}.$$

We bound the term $(B)$ using the finite element approximation bound in the output space $\mathcal{V}$ (Brenner & Scott, 2008, Eq. 4.4.28). First, we remark that by definition of $\hat{G}$ (see Fig. 6) and since $I_{\mathcal{U}} \circ P_{\mathcal{U}}(f) = P_{\mathcal{U}}(f)$, we have

$$\|\mathcal{G}(f_{h_1}) - \hat{\mathcal{G}}(f_{h_1})\|_{H^{s_2}(\Omega_2)} = \|\mathcal{G}(f_{h_1}) - P_{\mathcal{V}} \circ \mathcal{G}(f_{h_1})\|_{H^{s_2}(\Omega_2)}.$$

Then,

$$\|(I_d - P_{\mathcal{V}}) \circ \mathcal{G}(f_{h_1})\|_{H^{s_2}(\Omega_2)} \leq \|(I_d - P_{\mathcal{V}}) \circ \mathcal{G}(f) - (I_d - P_{\mathcal{V}}) \circ \mathcal{G}(f_{h_1})\|_{H^{s_2}(\Omega_2)} \\ + \|(I_d - P_{\mathcal{V}}) \circ \mathcal{G}(f)\|_{H^{s_2}(\Omega_2)}. \tag{14}$$

Since $I_d - P_{\mathcal{V}}$ is linear and continuous, $(I_d - P_{\mathcal{V}}) \circ \mathcal{G}$ is Lipschitz from $H^{s_1}(\Omega_1) \to \mathcal{V}_{h_2}$, therefore from $H^{s_1}(\Omega_1) \to H^{s_2}(\Omega_2)$ as $\mathcal{V}_{h_2} \subset \mathcal{V} \subset H^{s_2}(\Omega_2)$, and

$$\mathrm{Lip}((I_d - P_{\mathcal{V}}) \circ \mathcal{G}) \leq C_2 \mathrm{Lip}(\mathcal{G}).$$

Then, the first term in the right-hand side of Eq. (14) is bounded as

$$\|(I_d - P_{\mathcal{V}}) \circ \mathcal{G}(f) - (I_d - P_{\mathcal{V}}) \circ \mathcal{G}(f_{h_1})\|_{H^{s_2}(\Omega_2)} \leq C_2 \mathrm{Lip}(\mathcal{G})\|f - f_{h_1}\|_{H^{s_1}(\Omega_1)} \\ \leq C_1 C_2 \mathrm{Lip}(\mathcal{G}) h_1^{k_1 - s_1} \|f\|_{H^{k_1}(\Omega_1)},$$

using Brenner & Scott (2008, Eq. 4.4.28) on the input space $H^{s_1}(\Omega_1)$. We now bound the second term in Eq. (14) using the same finite element estimate on the output space $H^{s_2}(\Omega_2)$ as

$$\|(I_d - P_{\mathcal{V}}) \circ \mathcal{G}(f)\|_{H^{s_2}(\Omega_2)} \leq C_2 h_2^{k_2 - s_2} \|\mathcal{G}(f)\|_{H^{k_2}(\Omega_2)}.$$

Finally, the term $(C)$ is bounded as follows:

$$\|\hat{\mathcal{G}}(f_{h_1}) - \mathcal{S}_\theta(f_{h_1})\|_{H^{s_2}(\Omega_2)} = \|\hat{\mathcal{G}}(f_{h_1}) - \mathcal{D}_{\mathcal{V}_{h_2}} \circ \mathcal{P}_\theta \circ \mathcal{E}_{\mathcal{U}_{h_1}}(f_{h_1})\|_{H^{s_2}(\Omega_2)} \\ = \|\mathcal{D}_{\mathcal{V}_{h_2}} \circ (\mathcal{E}_{\mathcal{V}_{h_2}} \circ \hat{G} - \mathcal{P}_\theta \circ \mathcal{E}_{\mathcal{U}_{h_1}}) \circ f_{h_1}\|_{H^{s_2}(\Omega_2)} \\ \leq \|\mathcal{D}_{\mathcal{V}_{h_2}}\|_{\mathbb{R}^m \to H^{s_2}(\Omega_2)} \|(\mathcal{E}_{\mathcal{V}_{h_2}} \circ \hat{G} - \mathcal{P}_\theta \circ \mathcal{E}_{\mathcal{U}_{h_1}}) \circ f_{h_1}\|_{\ell^2(\mathbb{R}^m)},$$

as $\mathcal{D}_{\mathcal{V}_{h_2}} \circ \mathcal{E}_{\mathcal{V}_{h_2}}$ is the identity operator on $\mathcal{V}_{h_2}$ and using the submultiplicativity of the operator norm. First, we note that $\|\mathcal{D}_{\mathcal{V}_{h_2}}\|_{\mathbb{R}^m \to H^{s_2}(\Omega_2)} \leq \|\mathcal{D}_{\mathcal{V}_{h_2}}\|_{\mathbb{R}^m \to \mathcal{V}} \leq 1$. Additionally, using the fact that $\mathcal{D}_{\mathcal{U}_{h_1}} \circ \mathcal{E}_{\mathcal{U}_{h_1}}$ is the identity operator on $\mathcal{U}_{h_1}$, we obtain

$$\|\hat{\mathcal{G}}(f_{h_1}) - \mathcal{S}_\theta(f_{h_1})\|_{H^{s_2}(\Omega_2)} \leq \|(\mathcal{E}_{\mathcal{V}_{h_2}} \circ \hat{G} \circ \mathcal{D}_{\mathcal{U}_{h_1}} - \mathcal{P}_\theta) \circ \mathcal{E}_{\mathcal{U}_{h_1}}(f_{h_1})\|_{\ell^2(\mathbb{R}^m)}, \\ \leq \sup_{x \in K} \|\mathcal{E}_{\mathcal{V}_{h_2}} \circ \hat{G} \circ \mathcal{D}_{\mathcal{U}_{h_1}}(x) - \mathcal{P}_\theta(x)\|_{\ell^2(\mathbb{R}^m)} \leq \epsilon,$$

where the last inequality is due to Eq. (13). Combining the bounds for $(A)$, $(B)$, and $(C)$, we obtain the desired result. $\qquad\square$

## D ADDITIONAL DETAILS ON THE EXPERIMENTS

This section provides additional details on the numerical experiments presented in Section 4. All the experiments are conducted on a single *RTX 4070 Ti* GPU. For training, all the models are trained with the AdamW optimizer (Loshchilov et al., 2017) and using an exponential learning rate decay from $10^{-4}$ to $10^{-6}$ at the last epoch.

We consider two types of structure-preserving operator networks in these experiments. A single-level architecture that uses the message-passing model $\psi$, defined in Eq. (10), as processor, which we refer to as *SPON*, and a multigrid architecture that uses $\mathcal{P}_\theta^{MG}$ as processor, which we refer to as *SPON-MG*. The multigrid processor uses $\psi$ as its coarse model, as depicted in Fig. 2. We use the same architecture for all the message passing GNN layers. More specifically, we use two MLPs to compute the messages on each edge ($\phi_e$) and to update the node features ($\phi_v$), see Appendix B.2 for further detail.

### D.1 POISSON'S EQUATION WITH STRONG BOUNDARY CONDITIONS

For the Poisson experiment in Section 4.1, we consider a nonhomogeneous Dirichlet condition on the top boundary $\Gamma_D$ defined as $g = 10^{-2} \sin(\pi x)$, for $x \in \Gamma_D$. A $\text{CG}_1$ discretization is used for the input and output spaces, see Appendix A.1.

The source terms are generated using the Chebfun software system (Driscoll et al., 2014; Filip et al., 2019) as random smooth functions $f \sim \mathcal{GP}(0, K)$. Here, $K$ is a squared-exponential kernel with length-scale $0.4$. The corresponding solutions are generated by solving the Poisson problem, see Eq. (6), in Firedrake (Ham et al., 2023) and using LU decomposition.

The single-level structure-preserving operator network *SPON* uses the message-passing model $\psi$ as processor (cf. Eq. (10)). We consider a total number of 4 layers of message-passing layers for $\psi$, all with different parameters. For the multigrid architecture *SPON-MG*, we consider a hierarchy of $N = 3$ levels, with resolutions $n_x/4$, $n_x/2$, and $n_x$. We employ 1 message-passing layer for each level except for the coarse model $\psi$, for which we consider a total of 4 GNN layers. For this experiment, we consider a low-rank approximation of the weight matrices of $\psi$ for both *SPON* and *SPON-MG*, as detailed in Eq. (11). We use a compression factor $k$ of 20 for *SPON* and 1 for *SPON-MG*.

We train all the models for 500 epochs with a batch size of 4. For the loss, we use the $L^2$-relative error. It is worth noting that the computation of the $L^2$ norm is achieved using the finite element assembly, which is made possible by the fact that structure-preserving operator networks return finite element functions.

The Fourier neural operator is implemented using the `neuraloperator` (Li et al., 2024a) library with the default architecture settings for the Darcy flow datasets. Note that we do not use tensorization of the weights and add a zero padding to account for the non-periodic boundary conditions. The DeepONet implementation is based on the implementation of Lu et al. (2022; 2024a) and relies on the `DeepXDE` library (Lu et al., 2024b). DeepONets usually require a much larger number of training epochs than FNO, which might explain the poor performance of DeepONet on the Poisson benchmark in Fig. 3(a).

### D.2 FLUID FLOW PAST A CYLINDER

We consider a rectangular domain $[-5, 20] \times [-2.5, 2.5]$ with a circular obstacle of center $(0, 0)$ and radius $0.5$. We consider an unstructured mesh discretization comprising 9069 triangles, see top figure in Fig. 7.

To generate the dataset, we consider different inflow conditions on the left boundary. We generate 100 random source terms, sampled from a Gaussian process with periodic kernel (Williams & Rasmussen, 2006, Eq. 4.31), which we use for the left boundary. For each of them, we solve Eq. (7) using the finite element method to generate the corresponding trajectories, i.e. the fluid velocity $u(\cdot, t)$ for $t \in [0, T]$. In addition to the Dirichlet condition on the obstacle and the upper and lower sides, we also consider a compatibility "do-nothing" condition on $\Gamma_{out}$, the right side of $\Omega$, which is defined as $p\,\vec{n} = \frac{1}{\text{Re}} \nabla u \cdot \vec{n}$, where $\vec{n}$ denotes the outward normal vector, and $p$ is the pressure. The velocity and pressure are discretized using the Taylor–Hood finite element, i.e., continuous piecewise quadratic for the velocity and piecewise linear for the pressure, which is a stable and standard element pair for the Stokes equations (Taylor & Hood, 1973). We solve Eq. (7) using a backward Euler method up to $T = 30$ and with a timestep $\Delta t^{solve} = 0.1$. For each timestep, we solve the corresponding nonlinear problem using Newton's method with LU decomposition for the linear solver. We record the solution every 2 timestep, which results in 150 samples for each trajectory. We form the training dataset by considering the trajectories associated with 80 inflow conditions. The validation and test splits are formed by considering 10 trajectories for each split.

We train the *SPON-MG* model to learn the one-step forward operator $u(\cdot, t) \mapsto u(\cdot, t + \Delta t)$, for a relatively large prediction timestep $\Delta t = 2$. For training and inference, the predicted velocities are obtained autoregressively, i.e., by rolling out the predictions until the final timestep. The prediction timestep is 20 times greater than the timestep $\Delta t^{solve}$ used to solve Eq. (7) using FEM. To augment the amount of training data, for each trajectory (i.e. inflow condition), we also train our model using the subtrajectories between each prediction time step. More precisely, given that we generated the fluid velocities every $0.2s$ for each trajectory and that we consider a prediction timestep $\Delta t = 2s$, we

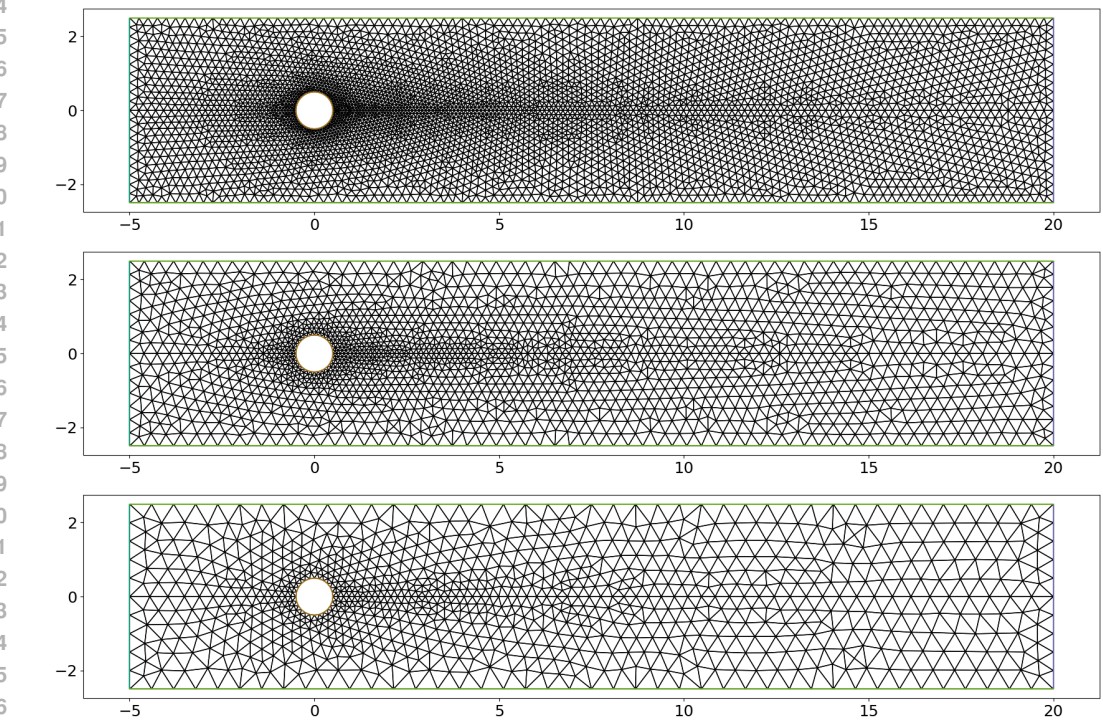

Figure 7: Mesh hierarchy considered for the Navier-Stokes problem in Section 4.2, where the coarse mesh (bottom), medium mesh (middle), and fine mesh (top) are composed of 1850, 3830, and 9069 triangles, respectively.

have 10 subtrajectories for each inflow condition. For example, the two first trajectories comprise the fluid velocities $u(\cdot, t)$ at times $t = 0, 2, 4, \ldots$, and $t = 0.2, 2.2, 4.2, \ldots$, respectively.

Table 2: Summary of the model for Navier–Stokes equation.

| Layer (type) | Output Shape | # Parameters |
|---|---|---|
| Encoder | [batch_size, 36848, 1] | – |
| MessagePassingMultiGridProcessor | [batch_size, 36848, 1] | – |
| 3 x MessagePassingBlocks | [batch_size, 36848, 1] | 3 x 337 |
| RestrictionFEM | [batch_size, 15692] | – |
| 3 x MessagePassingBlocks | [batch_size, 15692, 1] | 3 x 337 |
| RestrictionFEM | [batch_size, 7656] | – |
| *Coarse Model* | | |
| Linear | [batch_size, 500] | 3,828,000 |
| Linear | [batch_size, 7656] | 3,828,000 |
| 3 x MessagePassingBlocks | [batch_size, 7656, 1] | 3 x 337 |
| Linear | [batch_size, 500] | 3,828,000 |
| Linear | [batch_size, 7656] | 3,828,000 |
| ProlongationFEM | [batch_size, 15692] | – |
| Linear (smoother) | [batch_size, 15692, 1] | 3 |
| 3 x MessagePassingBlocks | [batch_size, 15692, 1] | (recursive) |
| ProlongationFEM | [batch_size, 36848] | – |
| Linear (smoother) | [batch_size, 36848, 1] | (recursive) |
| 3 x MessagePassingBlocks | [batch_size, 36848, 1] | (recursive) |
| Decoder | – | – |
| Total parameters: | | 15,315,036 |

For the multigrid processor, we consider a non-nested hierarchy of 3 unstructured meshes comprising 1850, 3830, and 9069 triangles, see Fig. 7. All the meshes are constructed using *Gmsh* (Geuzaine & Remacle, 2009). We consider 3 layers of message-passing GNNs for each level, all with different parameters. For the coarse model $\psi$, we consider a low-rank approximation with a compression factor $k$ of 500 to reduce the computational cost (cf. Eq. (11)), since the latent graph associated with the finest level contains approximately $40k$ nodes and $800k$ edges. A summary of the model is outlined in Table 2.

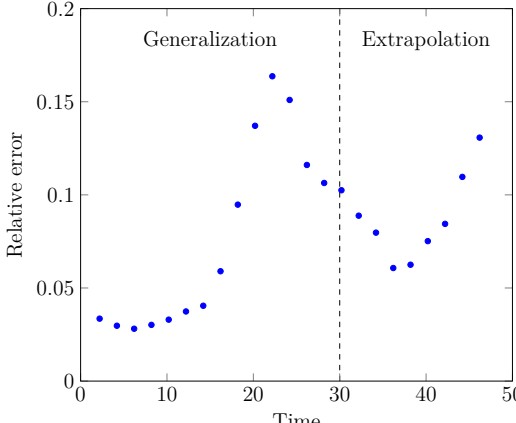

Figure 8: Relative error in the prediction of the velocity field for the Navier–Stokes problem, with the same initial condition as in Fig. 4. The left side of the figure displays the generalization capabilities of the model to new initial conditions, while the right side shows the extrapolation capabilities to new time steps, unseen during training.

We train our model for 15000 epochs with a batch size of 2. To improve generalization, we consider a divergence-free regularization on the loss function, as introduced by Bouziani et al. (2024), to enforce flow incompressibility on the model's predictions. Additionally, to reduce memory usage and training time, we unroll the full trajectory but only backpropagate through the last half of the predictions. This approach has similarities with the pushforward trick (Brandstetter et al., 2022), although the loss considered is different.

In Fig. 8, we report the generalization error, i.e., error across time steps that have been observed during training for a new source term in the test dataset, of our model for the source displayed in Fig. 4. Here, we observe a low $L^2$-relative error up to $t \approx 15$, where the fluid flow transitions from laminar to periodic behavior (see Fig. 9). Next in Figs. 8 and 9, we extrapolate the model further in time at time steps not seen during training (the model was trained up to $t = 30$) and observe a low relative error between the predicted velocity and ground truth, which demonstrates the model's ability to generalize to unseen time steps.

## E ADDITIONAL EXPERIMENTS

### E.1 HYPERELASTIC BEAM UNDER COMPRESSION

In this section, we consider a hyperelastic beam under compression on one of its boundary, and learn the mapping between the force applied on the boundary and the corresponding displacement field on the entire domain. This example illustrates how SPON architectures can be applied to problems where the input and output spaces are defined on different domains and with different spatial dimensions.

More specifically, we aim to approximate the solution operator $\mathcal{G} : H^1(\Gamma_R; \mathbb{R}_+) \to H^1(\Omega; \mathbb{R}^2)$ that maps the compression load $g$ to the beam displacement field $u$ satisfying the following nonlinear elasticity equation:

$$-\nabla \cdot P(u) = B \quad \text{in } \Omega, \tag{15}$$

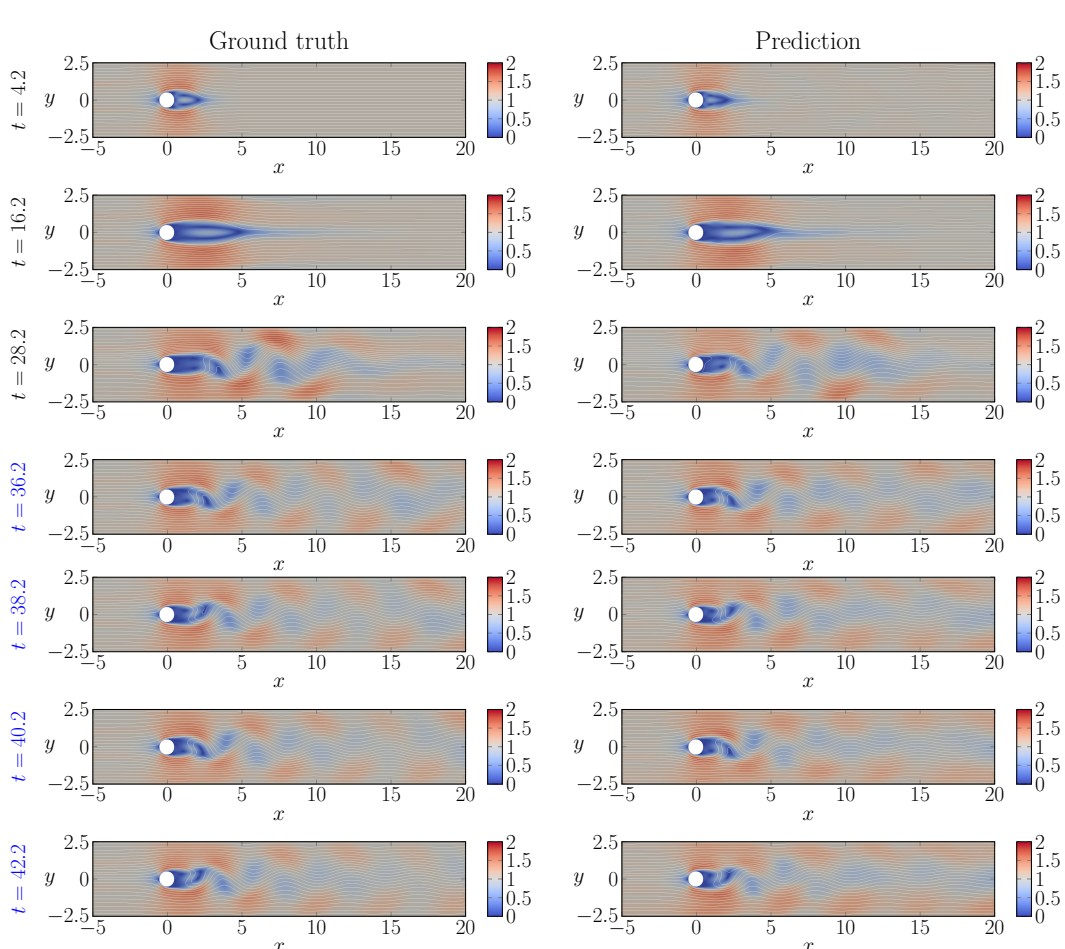

Figure 9: **Left:** Exact fluid velocity flow and magnitude from the random source (left boundary condition) in Fig. 4 at different time steps in the simulation. **Right:** Predicted velocity solution from the same initial condition at $t = 2.2$. The rows highlighted in blue are extrapolation tests, i.e., they correspond to time steps that have not been observed during training.

with $u_{|\Gamma_R} = (g, 0)^\top$, and where $\Omega = [0, 1] \times [0, 0.1]$ is the 2D computational domain, $\Gamma_R$ the 1D right boundary, $B = (0, -1000)^\top$, and $P(u)$ is the first Piola–Kirchhoff stress tensor given by

$$P(u) = \mu F(\text{tr}(C)I) - \mu F^\top + \frac{\lambda}{2}F^\top, \quad F = I + \nabla u, \quad C = F^\top F, \quad J = \det(F), \quad (16)$$

where the Lamé parameters $\mu(E, \nu) = E/(2(1+\nu))$ and $\lambda(E, \nu) = E\nu(1+\nu)(1-2\nu)$ are derived from Young's modulus $E = 10^6$ and Poisson ratio $\nu = 0.3$. We further complement Eq. (15) with the Dirichlet boundary condition $u(0, \cdot) = (0, 0)^\top$, along with natural boundary conditions $P(u) \cdot \vec{n} = 0$ on the top and bottom boundaries of $\Omega$, where $\vec{n}$ denotes the outward normal vector. The input and output spaces are discretized using $\text{CG}_2$ scalar and vector elements, respectively. For sake of simplicity, we consider constant load compressions on $\Gamma_R$, i.e. $g = -\epsilon$, with $\epsilon \in \mathbb{R}_+$.

The dataset is formed of $N = 80$ pairs of loads $\epsilon$, ranging from 0 to 0.2, and associated displacement $u$, solution to Eq. (15). The numerical solver consists of Newton's method with linesearch, along with GMRES for the linear solver. We use a continuation method and first solve for a load $\epsilon = 0.1$ and then use the solution as an initial guess for the next load. The dataset is split into 50 training samples, 20 validation samples, and 10 test samples.

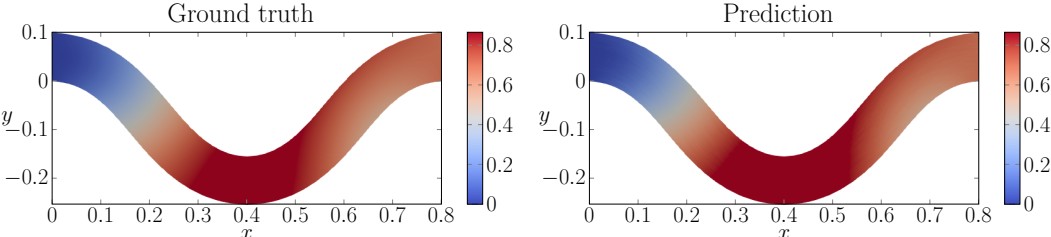

Figure 10: Ground truth displacement at $\epsilon = 0.1995$ against the prediction by the trained SPON.

We use a $40 \times 40$ rectangular grid for the beam, and an interval mesh with 40 points for $\Gamma_R$. We consider the single-level message passing processor *SPON* (with processor $\psi$). We train the model for 1000 epochs with a batch size of 4. The resulting SPON model achieves a relative $L^2$-error of $4.3 \times 10^{-2}$ on the test split and is capable of reproducing the hyperelastic deformations on the beam for larger values of compression load $\epsilon$ it was trained on, as shown in Fig. 10.

## E.2 Efficiency of the multigrid processor $\mathcal{P}_\theta^{MG}$

We have seen in Section 4.1 that the multigrid-based structure-preserving operator network (*SPON-MG*) surpasses the structure-preserving operator network that uses the single-level message-passing processor $\psi$ (*SPON*). In this section, we compare the efficiency of both architectures and compare how the number of parameters grow as we consider finer resolutions.

We consider the same setting as in Section 4.1, i.e., a unit square mesh of resolution $n_x$, and learn the solution operator associated with Eq. (6). Both architectures rely on the model $\psi$: *SPON* uses it at the finer level while it serves as the coarse model for *SPON-MG*, as illustrated in Fig. 2. As discussed in Appendix B.2, we employ a low-rank approximation for the weight matrices of $\psi$ (cf. Eq. (10)-Eq. (11)). For both SPONs, we use a compression factor $k$ of 5.

We report the evolution of the number of parameters and epoch time as we scale to higher resolution in Fig. 11. We can see in Fig. 11b that the number of parameters increases dramatically for the *SPON* model, reaching approximately $500M$ parameters for $n_x = 160$. This is due to the liner layers used in $\psi$, see Eq. (10), that largely dominate the total number of parameters. These linear layers comprise $\mathcal{O}(4 \times n_{DoFs}^2/k)$ parameters, with $k$ the compression factor, which equates $\mathcal{O}(4 \times n_x^4/k)$ since the degrees of freedom correspond to the grid nodes for a $\text{CG}_1$ discretization. For this experiment, we kept the same number of levels ($N = 3$) for *SPON-MG* as we move to finer resolutions. This explains why the number of parameters in Fig. 11b grows with the same rate but lead to significantly less parameters, since $\psi$ is, in this case, applied at the coarse level, i.e., at the resolution $n_x/4$. Alternatively, one could increase the number of levels for finer resolutions to reduce the number of parameters or even to keep it constant.

We also observe the substantial advantage of *SPON-MG* in terms of efficiency, as illustrated in Fig. 11a, with an epoch time that almost remains constant as we move from $n_x = 40$ to $160$, where the number of DoFs is 16 times greater. This contrasts with the single-level *SPON* architecture we considered for this experiment, for which the epoch time augments drastically, becoming 7 times slower than the *SPON-MG*. This speed-up results from delegating the computational load of message-passing updates and linear layers to the coarse level, where the number of degrees of freedom is lower.

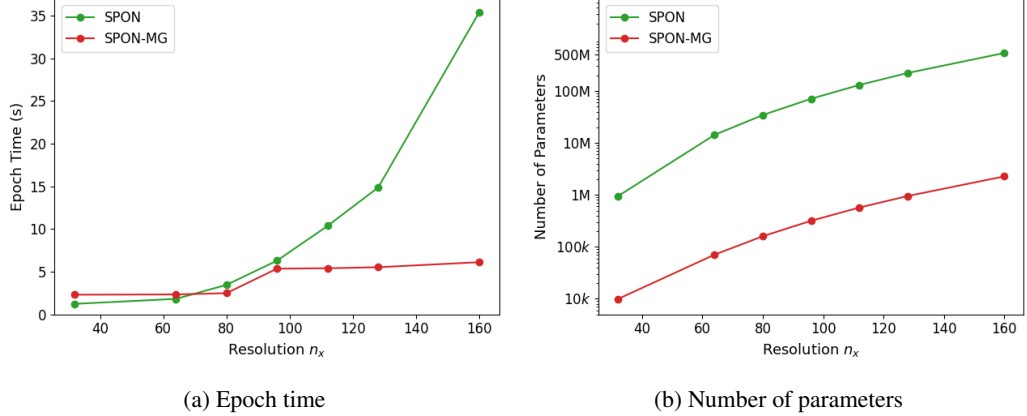

(a) Epoch time                    (b) Number of parameters

Figure 11: Comparison of the epoch time (left) and the number of parameters (right) for the single-level structure-preserving operator network (*SPON*) and the multigrid-based structure-preserving operator network (*SPON-MG*).

