# OpenReview forum: "Structure-Preserving Operator Learning"
_ICLR.cc/2025/Conference — ICLR 2025 Conference Withdrawn Submission_

### Official Review · Reviewer_DHEK · 2024-10-17

**Soundness:** 3
**Presentation:** 3
**Contribution:** 3
**Rating:** 6
**Confidence:** 4

**Summary:**

The authors present Structure-Preserving Operator Networks, a novel approach for learning operators. SPONs use FEM discretizations to preserve essential physical and mathematical properties, even for complex geometries. This framework ensures accurate enforcement of boundary conditions and provides theoretical guarantees. SPONs offer a balance between accuracy and computational efficiency. The authors also introduce a multigrid-inspired version of SPON for further performance gains and provide software tools to facilitate the design and training of these models.

**Strengths:**

1) The authors developed a new operator learning framework that maintains structure-preserving properties derived from finite element methods.
2) They established approximation bounds for a broad class of operators.
3) They conducted experiments on various PDEs, demonstrating the effectiveness of their approach.
4) They empirically demonstrated a trade-off between accuracy and efficiency.
5) The software is open-source and easy to use.
6) The paper is well-written, technically sound, and easy to follow.

**Weaknesses:**

1) There is a lack of baselines across all the experiments you conducted. For the Poisson equation, where the grid is uniform, one of the convolution-based architectures [1][2][3] could have been easily tested. In the other experiments, no baselines were included, despite the availability of numerous models that handle nonuniform discretizations [4][5]. While I understand that your goal isn't necessarily to outperform existing models, it would still be valuable for the community to see how your model compares to state-of-the-art approaches.

2) I believe the framework has not been tested extensively enough. There are many available datasets corresponding to various PDEs, such as the Wave equation, Compressible Euler equations, and others [6][2][3].

3) In most operator learning tasks, data is provided as pointwise values, typically generated using a numerical method. It appears that the FEM mesh used in your processor is closely tied to the mesh of the data. For example CG However, it's unclear how your method handles cases where the FEM mesh in the processor doesn't align with the discretization of the input, and how you obtain DOF in such situations.

[1] Ronneberger, Olaf, Philipp Fischer, and Thomas Brox. "U-net: Convolutional networks for biomedical image segmentation." Medical image computing and computer-assisted intervention–MICCAI 2015: 18th international conference, Munich, Germany, October 5-9, 2015, proceedings, part III 18. Springer International Publishing, 2015.

[2] Raonic, B., Molinaro, R., De Ryck, T., Rohner, T., Bartolucci, F., Alaifari, R., ... & de Bézenac, E. (2024). Convolutional neural operators for robust and accurate learning of PDEs. Advances in Neural Information Processing Systems, 36.

[3] Gupta, Jayesh K., and Johannes Brandstetter. "Towards multi-spatiotemporal-scale generalized pde modeling." arXiv preprint arXiv:2209.15616 (2022).

[4] Li, Z., Huang, D. Z., Liu, B., & Anandkumar, A. (2023). Fourier neural operator with learned deformations for pdes on general geometries. Journal of Machine Learning Research, 24(388), 1-26.

[5] Li, Z., Kovachki, N., Choy, C., Li, B., Kossaifi, J., Otta, S., ... & Anandkumar, A. (2024). Geometry-informed neural operator for large-scale 3d pdes. Advances in Neural Information Processing Systems, 36.

[6] Takamoto, M., Praditia, T., Leiteritz, R., MacKinlay, D., Alesiani, F., Pflüger, D., & Niepert, M. (2022). Pdebench: An extensive benchmark for scientific machine learning. Advances in Neural Information Processing Systems, 35, 1596-1611.

**Questions:**

1) Is there any advantage to using a higher-order mesh in the processor? For instance, when dealing with data on a uniform grid, what would be the benefit of using a CG2 mesh compared to a CG1 mesh?
2) What GPUs are used for training? How many samples for each benchmark are used for training?

**Details Of Ethics Concerns:**

/

---

### Official Review · Reviewer_v5dr · 2024-10-31

**Soundness:** 2
**Presentation:** 3
**Contribution:** 1
**Rating:** 3
**Confidence:** 4

**Summary:**

This paper presents the "Structure-Preserving Operator Learning" approach. Similar to other operator learning approaches presented in the literature, the Structure Preserving approach considers an Encoder-Processor-Decoder architecture. Unlike other typical approaches, it does not train the encoder and the decoder, but considers them as map from the space of input functions to the DOFs in the FEM space, and the opposite, respectively. Only the Processor, denoted as Structure-Preserving Operator Networks (SPONs), is trained in this case. The SPONs method is compared with FNO and DeepONets for one example and then two more are provided that compare the performance of SPON to the ground truth.

**Strengths:**

This paper provides a combination of operator learning with FEM and with multigrid methods. It is clearly written and easy to follow.

**Weaknesses:**

The paper suffers from multiple weaknesses:

a) There is no clear comparison to other methods. For example, the authors only compare against FNOs and DeepONets. DeepONets are not state-of-the-art and FNOs are not Representation Equivalent Neural Operator, as in Bartolucci et. al. that the authors cite. Therefore, in my understanding, the authors only compare for a very easy example, to methods that are bound to fail from the start, so comparison is not proper. I believe that the authors should compare to CNO [1], and then to the other methods that combine FEM and Operator Learning such as MINNs of Franco et. al. that the authors cite. Moreover, I am not sure why the authors do not make comparisons for the Flow around the Cylinder and the Beam under Compression.

[1] Raonic, B., Molinaro, R., De Ryck, T., Rohner, T., Bartolucci, F., Alaifari, R., Mishra, S. and de Bézenac, E., 2024. Convolutional neural operators for robust and accurate learning of PDEs. Advances in Neural Information Processing Systems, 36.

b) It is not clear to me what is the novelty and the contribution compared to Franco et. al. 2023 or other methods such as Lee et. al. 2023. These methods are referred to in the literature review, but it is not clear what their drawbacks are and how what the authors propose is different.

c) The authors make claims certain claims such tat SPONs have a  "Structural Property Preservation". Reading the paper, the authors repeat this multiple times, listing different properties each time. For example   line 49 "symmetries, boundary conditions, or conservation laws", line 21 "complex geometries", and also they write in line 179 "Other mathematical and physical properties may be satisfied at the discrete level using structure-preserving FE discretizations (Arnold et al., 2006)." However, they do not provide examples to showcase these properties, or an example that other operator learning methods fail to preserve this structure while their methods does not.

d) It is not clear to me what is the advantage of SPONs compared to traditional solvers. The experiments that the authors present are rather easy to solve with open source packages.

**Questions:**

Could you please provide more information on what is the difference between the method you propose and Franco et. al. 2023 and Lee et. al. 2023?

How does SPON compare to Multipole Neural Operators [1]?

Li, Z., Kovachki, N., Azizzadenesheli, K., Liu, B., Stuart, A., Bhattacharya, K. and Anandkumar, A., 2020. Multipole graph neural operator for parametric partial differential equations. Advances in Neural Information Processing Systems, 33, pp.6755-6766.

---

### Official Review · Reviewer_f3n4 · 2024-11-02

**Soundness:** 2
**Presentation:** 3
**Contribution:** 3
**Rating:** 5
**Confidence:** 3

**Summary:**

The paper introduces Structure-Preserving Operator Networks, a framework for learning operators from data that preserves key properties of continuous systems using FEM discretizations. SPONs are differentiable and can handle complex geometries and boundary conditions. The paper also presents a multigrid-inspired SPON architecture for improved efficiency and reduced parameters.

**Strengths:**

1. The paper is well-written and well-organized, making it easy to understand.

2. The idea is novel. It addresses significant issues relevant to the AI4PDE field, highlighting its importance and potential applications.

3. The proposed method can automatically satisfy the boundary conditions.

4. The author provides several theoretical insights.

**Weaknesses:**

The main issue with this paper lies in its experimental design. I listed my major concerns as follows:

1. The authors conducted only two experiments, one on the Poisson equation and one on the flow around a cylinder. This limited experiments may affect the generalizability and reliability of the proposed methods.

2. For the Poisson equation experiment, the authors only compared their method with basic approaches (like FNO and DeepONet), which are three years old, without including more advanced methods, such as F-FNO. In the cylinder flow problem, there was no comparison with any other methods, limiting the comprehensive evaluation of their performance.

3. In the cylinder flow experiment, the authors did not report any quantitative metrics to assess model performance, making it difficult to evaluate the effectiveness of their approach.

4. The paper does not demonstrate how the neural network accelerates performance compared to traditional methods, particularly in achieving the same accuracy. This comparison is crucial for highlighting the advantages of their approach.

5. While the authors mention methods that use GNN simulators in their related work part, they do not experimentally validate the superiority of their method over these approaches.

**Questions:**

Q1: Why do the two graphs in Fig. 1 have a different number of nodes?

Q2: How is the mesh generated, and how does the author plan to handle more complex geometric problems? As far as I know, traditional finite element methods require significant effort to create the mesh.

---

### Official Review · Reviewer_RxcH · 2024-11-03

**Soundness:** 2
**Presentation:** 2
**Contribution:** 2
**Rating:** 3
**Confidence:** 4

**Summary:**

The authors proposed SPON, a method to integrate neural message passing PDE solvers into FEM framework. FEM basis functions are used to encode and decode the input and output of message passing neural networks. The authors also enhanced their model by introducing a multigrid-based processor network. Experimental results on 2D Poisson equation and 2D cylinder flow demonstrate accuracy improvements comparing to previous works.

**Strengths:**

1. The authors introduced the FEM framework into neural PDE solvers, which generalized previous works that formulate the problem as a discrete mapping. This formulation is well-suited for problems on irregular domains. Notably, this formulation enables discretization-invariance in GNN architectures.

2. The proposed method is enhanced with multigrid methods, which improves the performance and efficiency of the network.

**Weaknesses:**

1. Lack of comparison with previous GNN-based solvers, both methodologically and experimentally. Based on my understanding, SPON (without -MG) differs from GNN solvers only on its encoder and decoder, i.e., a different input/output embedding scheme. It is not clear whether the proposed method still brings improvement, in terms of both accuracy and efficiency, over these GNNs.

2. Soundness of comparison. The authors compared their model to FNOs on 2D Poisson equation with Dirichlet BC, and reported that FNO achieves 1.71% relative error on the test set. It is known that these spectral-based methods have very strong expressive power, especially for problems on regular domains. FNO is expected to be able to parameterize the (linear) solution operator within one spectral convolution layer.

3. Minor writing and presentation issues. See question 2 below.

4. No efficiency and accuracy comparison with numerical solvers are provided in the cylinder flow example.

**Questions:**

1. How does SPON compare to GNN solvers?

2. In equation 1, I believe the composition notation $\circ$ is misused. In figure 1, it is not clear to me what the colored dots mean, and why node connectivity changes. In equation 6, why is $f$ in $H^1$ instead of the dual space $H^{-1}$. In equation 7, $f$ is undefined.

---

### Note · Authors · 2024-12-10

I have read and agree with the venue's withdrawal policy on behalf of myself and my co-authors.